# Populus simonii × Populus nigra overexpressing PsnWRKY70 recruits phyllosphere bacterial strains that inhibit Alternaria alternata

Wei Wang,[1,2,3] Weixiong Wang,[1,4,5] Jing Jiang,[1,5] Xiangdong Bai,[1,6] Kun Chen,[1,7] Xiaoyue Zhang,[1,7] Jingya Yang,[1,4,5] Di Wu,[1,4,5] Ben Niu,[1,4,5] Guifeng Liu[1,5]

**ABSTRACT**  WRKY transcription factors have been implicated in the regulation of disease resistance associated with plant immune responses, which has crucial implications for defense responses against stress in plants. The role played by the *PsnWRKY70* gene of *Populus* (*Populus simonii* × *P. nigra*) in triggering the mechanism between the phyllosphere microbiome and plant defense against foliar pathogens remains unclear. Molecular ecological network analysis demonstrated that the stability and complexity of the phyllosphere bacterial community of *Populus* were influenced by *Alternaria alternata* infection. Specifically, compared to the wild-type line, the *PsnWRKY70*-overexpressing (OE) line had a higher average clustering coefficient and modularity. Furthermore, metabolomic analysis revealed that 19 differential metabolites were significantly enriched in the leaves of the OE line. Among these metabolites, coumarin compounds, such as fraxetin-8-O-glucoside (fraxin) and scopoletin-7-O-glucoside (scopolin), significantly promoted the proliferation of the genera *Methylobacterium* and *Achromobacter* with resistance to *A. alternata*. Additionally, these genera also served as connectors in the molecular ecological network of the phyllosphere microbiome of the OE line. Thus, we concluded that the *PsnWRKY70* gene enhanced the stability, complexity, and core taxa cooperation of the phyllosphere microbial network in *Populus* and regulated the biosynthesis of fraxin and scopolin to recruit beneficial bacteria controlling *A. alternata* infection. These findings provide valuable insights into the ability of resistant plant genotypes to drive the assembly of the phyllosphere microbiome, advancing our understanding of defense against pathogens using the biocontrol phyllosphere microbial community.

**IMPORTANCE**  Poplar leaf blight caused by *Alternaria alternata*, a common disease in Northeast China, can cause abnormal abscission of poplar leaves and even lead to plant death in severe cases. WRKY transcription factors have been implicated in the regulation of disease resistance associated with plant immune responses to secondary metabolism via a complicated gene network. However, little is known about how the metabolites regulated by the *PsnWRKY70* gene trigger changes in the phyllosphere microbiome, leading to increased resistance to foliar pathogens. Here, the *PsnWRKY70* overexpressing line of *Populus* (*Populus simonii* × *P. nigra*) exhibited increased coumarin synthesis in the leaves, triggering changes in microbial species central in phyllosphere microbial networks and leading to increased resistance to *A. alternata* infection. This study provides insights into the role of the *PsnWRKY70* gene in triggering the resistance mechanism to *A. alternata* in *Populus*.

**KEYWORDS**  *Populus simonii* × *P. nigra*, *PsnWRKY70*, phyllosphere bacterial community, coumarin, *Alternaria alternata*

Address correspondence to Di Wu, wudi_nefu@nefu.edu.cn, Ben Niu, ben_niu@nefu.edu.cn, or Guifeng Liu, liuguifeng@126.com.

Wei Wang and Weixiong Wang contributed equally to this article. Author order was determined by drawing straws.

The authors declare no conflict of interest.

See the funding table on p. 17.

As an important part of the plant microbiome (1, 2), phyllosphere microbes play key roles in plant growth and performance by mediating nutrient absorption (3), secreting phytohormones, and increasing tolerance to environmental stress. In addition, the microbial communities on phyllosphere surfaces have the potential to maintain plant health through competition, antagonism, and induction of plant defense responses to suppress infection by foliar phytopathogens (4–7). Since the phyllosphere is an open niche, the composition of its microbial communities is highly sensitive and often regulated by factors such as the host genotype, plant species, and environmental conditions (8–10).

As an important determinant of the colonization and structure of microbial communities in the phyllosphere, plant genetic networks can drive microbial immigration and emigration by mediating leaf structure, leaf exudates, and volatiles (11–15). Plant genetic networks can shape phyllosphere microbial communities via their immune system and maintain plant health (16). Impaired plant immunity networks can alter the microbial composition and cause dysbiosis in the phyllosphere (13), triggering leaf diseases. For example, defective in pattern-triggered immunity (PTI) and cell surface component structuring genes, the *Arabidopsis* quadruple mutant (*min7 fls2 efr cerk1*) builds a less diverse bacterial community in the leaf endosphere, resulting in leaf disease symptoms related to dysbiosis (14). *Populus simonii* × *Populus nigra* is widely distributed throughout Northern China with wide adaptability, rapid growth, and strong resistance and can be used for landscaping, afforestation, and industrial materials (17, 18). However, how immunity networks in *Populus* shape the assembly of the phyllosphere-associated microbiota for the defense of plant leaf health remains largely unknown.

WRKY, a family of transcription factors (TFs) in plants, has diverse biological functions, including plant growth and development, plant immune system responses to various biotic and abiotic stressors, secondary metabolism, and hormone signal transduction via a complicated gene network (19–22). Many studies have shown that WRKYs are crucial regulators of plant defense responses to pathogen infection (19, 23). Among them, *WRKY70* plays an important role in plant defense against pathogen infection by activating genes related to microbe- or pathogen-associated molecular PTI and effector-triggered immunity (ETI) (24, 25). Transgenic wheat lines overexpressing (OE) *HvWRKY70* have shown significantly stronger resistance to *Blumeria graminis* f. sp. *tritici* pathotype E20 and *Puccinia striiformis* f. sp. *tritici* pathotype *CYR32* compared with wild-type (WT) plants (26). In previous research, we found that the transcription factor *PsnWRKY70* was involved in regulating the composition of the poplar phyllosphere microbiome. It recruited beneficial microbes leading to enhanced poplar resistance to *Melampsora laricis*-populina (27). We also found that the *PsnWRKY70*-OE line specifically enriched the genera *Paracoccus*, *Arthrobacter*, and *Rhodococcus* after the leaves of poplar seedlings generated via aseptic tissue culture were inoculated with a consortium consisting of 39 bacterial strains, compared to the wild-type line (28). Furthermore, WRKY TFs can regulate the biosynthesis of numerous secondary metabolites, such as alkaloids, phenylpropanoids, terpenes, anthocyanidins, and phytoalexins, to defend plants against pathogen infection (29). Although plant immunity networks are known to affect the phyllosphere microbiome, it remains unknown whether WRKY70 TFs play a role in shaping the microbial community structure in the phyllosphere, leading to limited pathogen infection via the regulation of the secondary metabolites produced by *Populus*.

Poplar leaf blight caused by *Alternaria alternata* is a common disease in Northeast China and is more epidemic in nurseries (30). Leaf blight disease causes abnormal abscission of poplar leaves and can lead to plant death in severe cases (31). It has been confirmed that *PsnWRKY70* enhances poplar's defense against *A. alternata* by activating genes in both PTI and ETI (24). However, no studies have combined leaf metabolomics and phyllosphere bacterial microbiome to better understand the mechanism by which *PsnWRKY70* regulates *Populus* (*Populus simonii* × *P. nigra*), triggering resistance to *A. alternata* infection. This study aimed to: (i) determine the distribution pattern and

diversity of phyllosphere bacterial communities between *PsnWRKY70*-OE and WT lines of *Populus*, (ii) determine the relationships among potential key functional taxa within phyllosphere bacterial communities and the key specific metabolites regulated by the *PsnWRKY70* gene, and (iii) uncover the mechanism by which key specific metabolites regulated by the *PsnWRKY70* gene in leaves recruit the assembly of key phyllosphere biocontrol bacterial species. In this study, leaf metabolites and the phyllosphere bacterial microbiome of OE and WT lines were profiled using a widely targeted metabolomics technique and 16S ribosomal gene amplicon sequencing. This study provides new insight into the *PsnWRKY70* gene's contribution to disease resistance against *A. alternata* by altering leaf metabolite patterns, promoting the development of new strategies for woody plant disease resistance, and revealing interactions between plants and their associated microbiota.

## RESULTS

### Expression analysis of *PsnWRKY70* in *Populus* leaves

To confirm the impact of non-inoculation and inoculation with *A. alternata* on the expression of the *PsnWRKY70* gene in the WT and *PsnWRKY70*-OE lines of *Populus*, the expression levels of the *PsnWRKY70* gene in these two lines were detected by Quantitative reverse transcription polymerase chain reaction (qRT-PCR). The expression level of the *PsnWRKY70* gene in the OE line was significantly higher than that in the WT line during the period from 0 to 36 h, regardless of inoculation with *A. alternata* (Fig. S1A and B). When not inoculated with *A. alternata*, the expression level of the *PsnWRKY70* gene in the OE line was 1.56–1.90 times that of the WT line (Fig. S1A). After inoculation with *A. alternata*, the expression level of the *PsnWRKY70* gene in the OE line was 1.98–16.23 times that of the WT line (Fig. S1B). There was a significant increase in the expression of the *PsnWRKY70* gene in the OE line compared to WT after inoculation with *A. alternata*. This indicates that *A. alternata* infection could significantly upregulate the *PsnWRKY70* expression level in the OE line of *Populus*.

### Taxonomic characteristics of phyllosphere microbial communities

In this study, the WT line of *Populus* infected by *A. alternata* was more susceptible to symptoms of leaf blight than the OE line (Fig. S2). To test whether the *PsnWRKY70* modulates leaf microbiome assembly, thereby limiting pathogen development, we compared the structure and diversity of the phyllosphere bacterial community from the WT and OE lines of *Populus* that were non-inoculated or inoculated with *A. alternata*. A total of 406,854 high-quality bacterial reads were obtained from 39 phyllosphere microbial samples from OE and WT lines of *Populus* and clustered into 4,102 amplicon sequence variants (ASVs; Table S1).

The identified sequences of the phyllosphere bacterial community were affiliated with 23 bacterial phyla. The phyla Proteobacteria, Actinobacteria, and Firmicutes were dominant, followed by Bacteroides, in all phyllosphere bacterial communities, accounting for more than 86.19% of the total reads (Table S2). The composition of phyllosphere bacterial communities in the OE line not inoculated with *A. alternata* was similar to that in the WT line, although the abundance levels were weakly different (Fig. 1A). However, there was a significant change in the phyllosphere community between OE and WT lines after infection with *A. alternata* (Fig. 1A). A significantly higher abundance of the phyla Proteobacteria and Bacteroides but a significantly lower abundance of the phylum Actinobacteria characterized the phyllosphere bacterial community from the OE line, while those in the WT line had a lower abundance of the phylum Proteobacteria and a higher abundance of the phylum Actinobacteria (Fig. 1 and Fig. S3A). Compared to non-inoculated plants, the relative abundances of the phylum Firmicutes and Thermi in the phyllosphere bacterial community from the WT line declined significantly after infection with *A. alternata*, while a significantly higher abundance of phylum Proteobacteria was observed along with a significantly lower abundance of the phyla Actinobacteria and Firmicutes in the OE line (Fig. S3B and C).

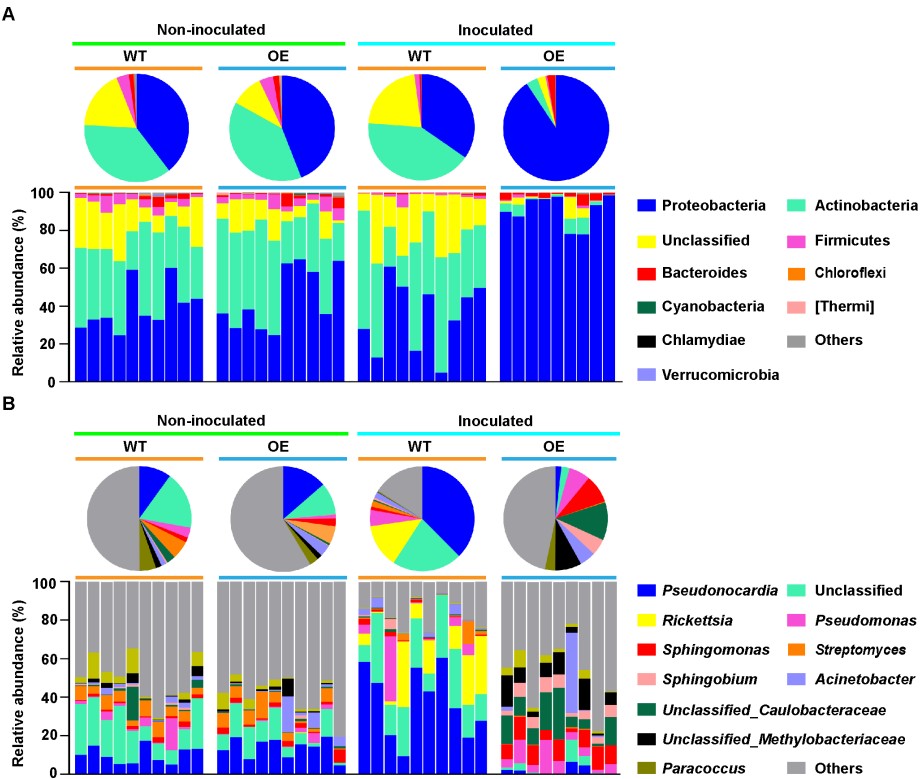

**FIG 1** Taxonomic composition of phyllosphere microbial communities between *PsnWRKY70*-OE and WT lines of *Populus* at the phylum (A) and genus level (B) in non-inoculated and inoculated with *A. alternata*.

To further identify specific differences in phyllosphere microbial communities between the WT and OE lines, we analyzed the microbial community structures at the genus level (Fig. 1B). The high-quality phyllosphere bacterial sequences from all samples were assigned to 236 identified genera. Of these, *Pseudonocardia* (16.1%), *Pseudomonas* (3.94%), *Rickettsia* (3.48%), *Streptomyces* (3.41%), *Sphingomonas* (3.3%), *Acinetobacter* (2.94%), *Paracoccus* (2.88%), and *Sphingobium* (1.52%) were the most abundant genera present in all samples (defined as having an average relative abundance >1%; Table S3). A significantly higher abundance of the genera *Sphingomonas*, *Paracoccus*, and *Sphingobium* characterized the phyllosphere bacterial community from the OE line inoculated with *A. alternata*, while those in the WT line only had a higher abundance of the genera *Pseudonocardia* and *Rickettsia* (Fig. 1 and Fig. S4A). After inoculation with *A. alternata*, the relative abundances of the genera *Pseudonocardia* and *Rickettsia* in the phyllosphere bacterial community from the WT line showed an obvious increase, compared to non-inoculated plants (Fig. S4B). In the OE line, significantly higher abundances of the genera *Pseudomonas* and *Sphingobium* were observed in samples inoculated with *A. alternata* compared to non-inoculated samples (Fig. S4C). This indicates that the *PsnWRKY70* gene plays a more significant role in shaping the community composition of the phyllosphere microbiomes of *Populus* when infected with *A. alternata*.

## Differences in the diversity of phyllosphere microbial communities

Alpha diversity analysis (Observed species, Shannon index, Evenness, and Simpson index) was used to determine whether the plant genotype affected the diversity and richness of the phyllosphere microbial community in *Populus*. Significant differences were observed in the Evenness and Simpson indices of the phyllosphere microbial community between non-inoculated the OE and WT lines of *Populus* (Fig. 2A). However, the four indices in the phyllosphere microbial community of the OE line were significantly higher than those in the WT line after inoculation with *A. alternata* (Fig.

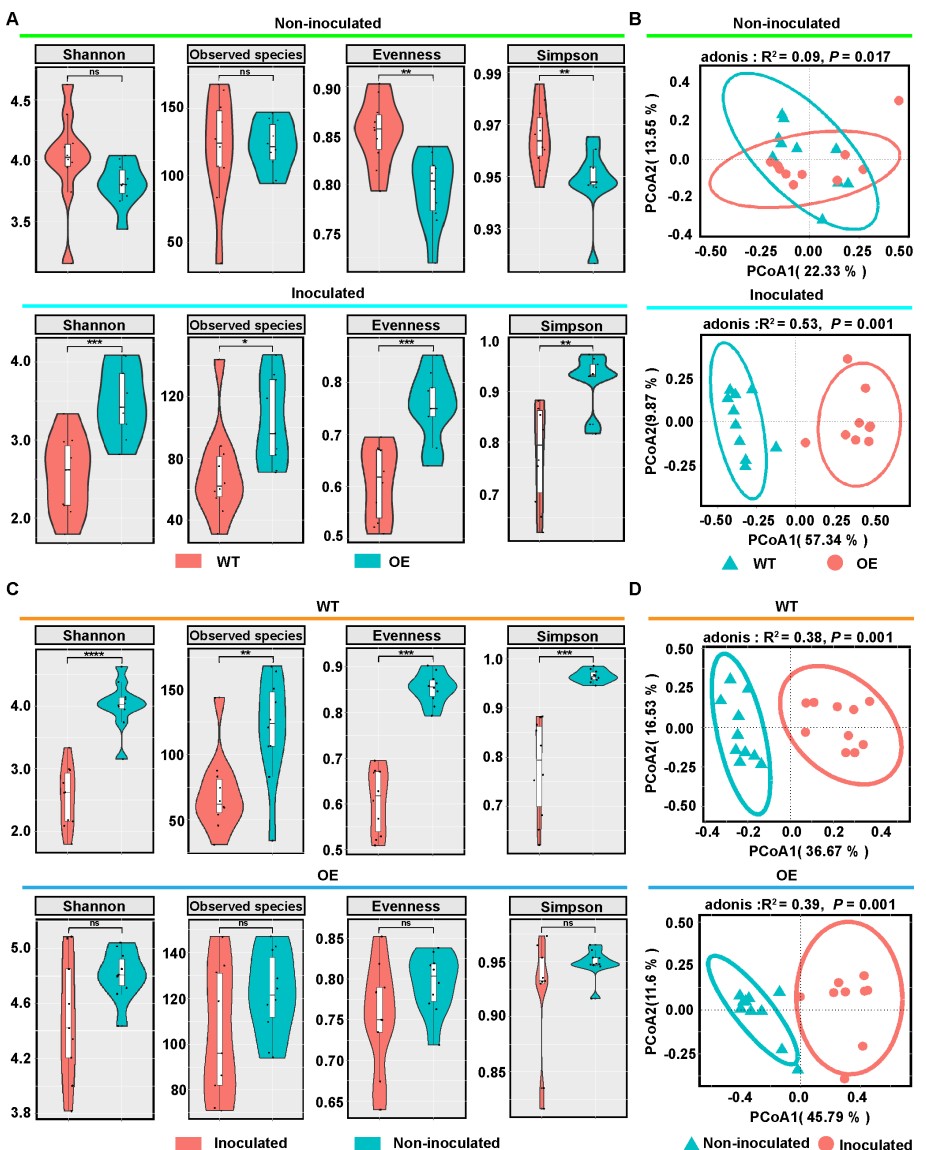

**FIG 2** Alpha and principal coordinates analysis (PCoA) of phyllosphere microbial communities between *PsnWRKY70*-OE and WT lines of *Populus*. (A) Four diversity indices and significant difference analysis of phyllosphere microbial communities among *Populus* genotypes that were non-inoculated or inoculated with *A. alternata*. *$P < 0.05$; **$P < 0.01$; ***$P < 0.001$; ns indicates no significant differences. (B) PCoA of the phyllosphere microbial communities between the WT and OE lines that were non-inoculated or inoculated with *A. alternata*. (C) Four diversity indices and significant difference analysis of phyllosphere microbial communities between WT or OE lines that were non-inoculated and inoculated with *A. alternata*, respectively. *$P < 0.05$; **$P < 0.01$; ***$P < 0.001$; ns indicates no significant differences. (D) PCoA of the phyllosphere microbial communities in WT or OE lines that were non-inoculated and inoculated with *A. alternata*.

2A). There were no significant differences in the α-diversity index in the phyllosphere microbial community of the OE line between the *A. alternata*-inoculated and non-inoculated treatments; however, the four indices were significantly decreased in the WT line inoculated with *A. alternata* compared to the non-inoculated WT line (Fig. 2C). These results indicate that the *PsnWRKY70* gene in *Populus* has a stronger effect on the diversity, richness, and evenness of the phyllosphere bacterial community when infected by a pathogen.

Principal coordinate analysis (PCoA) was used to analyze the overall structural variations in phyllosphere bacterial communities between OE and WT lines. The

**TABLE 1** Topological characteristics of the molecular ecological networks of the phyllosphere microbial community of *Populus*

| Communities | | Non-inoculated | | Inoculated | |
|---|---|---|---|---|---|
| | | WT | OE | WT | OE |
| Empirical networks | Total nodes | 104 | 101 | 44 | 77 |
| | Total links | 678 | 579 | 67 | 170 |
| | $R^2$ of power-law | 0.298 | 0.504 | 0.789 | 0.596 |
| | Average degree | 13.038 | 11.465 | 3.045 | 4.416 |
| | Average path distance | 2.232 | 2.229 | 2.987 | 3.135 |
| | Average clustering coefficient | 0.130 | 0.164 | 0.014 | 0.083 |
| | Modularity | 0.234 | 0.264 | 0.407 | 0.531 |
| Random networks | Average path distance | 2.146 ± 0.013 | 2.200 ± 0.016 | 3.042 ± 0.134 | 2.953 ± 0.080 |
| | Average clustering coefficient | 0.271 ± 0.015 | 0.224 ± 0.012 | 0.096 ± 0.029 | 0.118 ± 0.024 |
| | Modularity | 0.178 ± 0.008 | 0.200 ± 0.007 | 0.458 ± 0.019 | 0.3834 ± 0.012 |

phyllosphere microbial communities were significantly segregated between the two *Populus* genotypes (Fig. 2B). Permutational multivariate analysis of variance (PERMA-NOVA) analysis revealed a stronger distinction in phyllosphere microbial communities between OE and WT lines that were inoculated ($R^2 = 0.53$, $P = 0.001$) with *A. alternata*, and a weaker distinction between the non-inoculated OE and WT lines ($R^2 = 0.09$, $P = 0.017$; Fig. 2B). The phyllosphere bacterial community was more influenced by the host genotype after inoculation with *A. alternata*, compared to the non-inoculated plants. A clear separation of bacterial communities was observed across the first principal coordinate. Furthermore, the phyllosphere microbial communities of both WT ($R^2 = 0.38$, $P = 0.001$) and OE ($R^2 = 0.39$, $P = 0.001$) lines inoculated with *A. alternata* showed significant deviations along the positive axis of the first principal coordinate, compared to the non-inoculated WT and OE lines (Fig. 2C and D). In summary, the bacterial community composition of the phyllosphere microbiome significantly differed between the two *Populus* genotypes.

## Molecular ecological networks of the phyllosphere microbiome

To evaluate the influence of the different *Populus* genotypes on the molecular ecological patterns in the phyllosphere microbiome, we constructed random matrix theory (RMT)-based networks. The molecular ecological networks of the WT and OE lines inoculated with *A. alternata* were characterized by a lower number of total nodes and links and a lower average degree, but a higher $R^2$ of the power-law, average path distance, and modularity, compared to non-inoculated lines. The empirical networks of the phyllosphere microbiome from two genotypes for both the *A. alternata*-inoculated and non-inoculated treatments had a greater average path distance and higher modularity than their corresponding randomized networks, indicating small-world behavior and modular features (32). However, the empirical network from the WT line inoculated with *A. alternata* had lower modularity compared to that of the non-inoculated line, suggesting that the modular feature was destroyed. Multiple network topological metrics showed that the co-occurrence networks of phyllosphere microbiomes showed similar patterns between two *Populus* genotypes not inoculated with *A. alternata*, while different bacterial molecular ecological patterns were observed between the OE and WT lines after inoculation with *A. alternata* (Table 1). The bacterial network of the OE line inoculated with *A. alternata* had the highest number of total nodes and links, the highest average path distance, and greater average connectivity (avgK) (Table 1 and Fig. 3), indicating that it has a more complex network, greater network robustness, and higher density of connections compared to that of the WT line. Additionally, the network of the OE line had a higher average clustering coefficient, modularity, and positive correlation. These results indicate that the potential cooperative behaviors (positive correlations) of bacterial networks in OE increased significantly compared to those of the phyllosphere microbiome for the WT line after inoculation with *A. alternata* (Fig. 3).

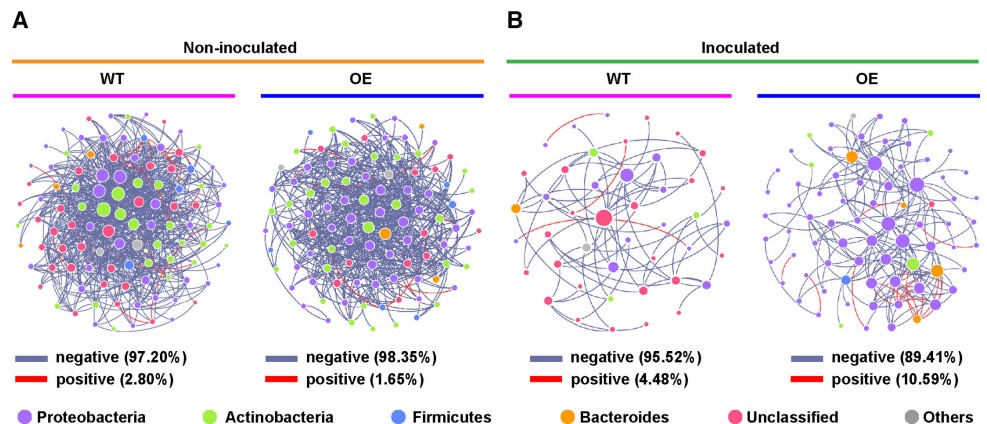

**FIG 3** Co-occurrence networks of phyllosphere microbial communities in *PsnWRKY70*-OE and WT *Populus* lines non-inoculated (A) or inoculated (B) with *A. alternata*. Each node represents an ASV, and the same-colored nodes indicate the same phylum. The size of these ASVs was proportional to the relative abundance of the corresponding ASV. The line color represents a positive (pink) or negative (gray) correlation.

To identify the keystone taxa in the bacterial network of the phyllosphere microbiomes, all ASVs from OE and WT lines were evaluated using within-module connectivity ($Zi$) and among-module connectivity ($Pi$). Most of the keystone taxa in the network structure of the *Populus* phyllosphere microbiomes were classified as connectors when not inoculated with *A. alternata*. A total of 59 connectors (24 ASVs in the WT line and 35 ASVs in the OE line) in the bacterial network belonged to the phyla Proteobacteria, Actinobacteria, Bacteroidetes, Chloroflexi, Thermi, and unclassified Bacteria (Table S4). After inoculation with *A. alternata*, the number of keystone taxa significantly decreased in the bacterial network from the WT and OE lines. One module hub and three connectors were detected in the OE network as keystone nodes belonging to the phylum Proteobacteria, while only one ASV (unclassified Bacteria) was classified as a module hub for WT networks (Table S5). A network hub was not detected in any of the bacterial networks of the phyllosphere microbiome. Overall, the number of keystone taxa in the network structure of the phyllosphere microbiome of the OE line was greater than that in the WT line, regardless of inoculation with *A. alternata*.

## Antagonistic potential of phyllosphere bacterial communities against *A. alternata*

Based on linear discriminant analysis effect size (LEfSe), potential discriminative taxa of phyllosphere microbial communities between OE and WT lines were further distinguished. LEfSe analysis identified eight and four discriminative taxa in the phyllosphere microbiome in OE and WT lines not inoculated with *A. alternata*, respectively (Fig. 4A). The number of discriminative taxa for the OE line significantly increased (29) after inoculation with *A. alternata* (Fig. 4A and Table S11).

A total of 567 phyllosphere bacteria strains were isolated and affiliated with four phyla (Proteobacteria, Firmicutes, Actinobacteria, and Bacteroidetes) and 68 genera (Table S8). Among them, 31 identified genera were detected in the community composition of the phyllosphere microbiome at the genus level. The relative abundance of these 31 genera accounted for 16.28% of the total relative abundance of microbial species in the phyllosphere microbiome of *Populus* (Table S9). Among the 567 phyllosphere bacteria isolates, the genera *Microbacterium, Arthrobacter,* and *Frigoribacterium* had relatively high proportions, accounting for 20.63%, 13.40%, and 9.35%, respectively (Table S8).

Plant resistance genes can enhance the plant resistance to pathogen infection by influencing the assembly of the plant's beneficial microbiome (33). Considering that a stronger distinction in phyllosphere microbial communities between OE and WT lines inoculated with *A. alternata*, the significantly discriminative genera enriched in the OE

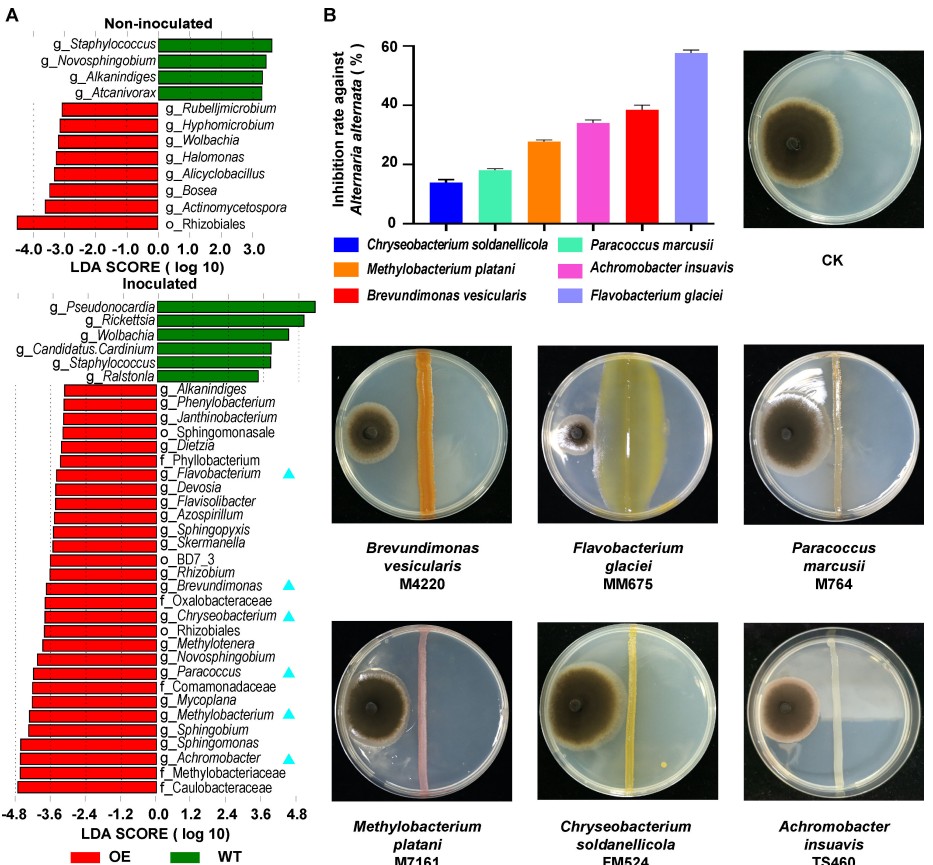

**FIG 4** LEfSe analysis of phyllosphere community composition and isolation of bacterial strains with antagonistic potential against *A. alternata*. (A) Discriminant analysis of species differences in phyllosphere community composition between *PsnWRKY70*-OE and WT *Populus* lines non-inoculated or inoculated with *A. alternata* based on LEfSe analysis. Six significantly discriminative genera enriched in the OE line were selected for the antagonistic bioassays against *A. alternata* and were marked with blue triangles. (B) *In vitro* antagonism bioassays against *A. alternata* and the inhibition rate of six antagonistic isolates against *A. alternata*.

line inoculated with *A. alternata* were used as a reference to screen for antagonistic strains (Fig. 4A). Among these isolates, 18 isolates, belonging to the genera *Brevundimonas* (8), *Methylobacterium* (1), *Paracoccus* (2), *Chryseobacterium* (4), *Flavobacterium* (1), and *Achromobacter* (2), were identical to the significantly enriched discriminative genera enriched in the OE line inoculated with *A. alternata* (Table S8 and S11). Among these six genera, bacterial isolates exhibiting the highest similarity to the ASVs with the highest relative abundance in each genus of the phyllosphere microbial communities of *Populus* were selected for the antagonistic bioassays against *A. alternata* (Table S12). Therefore, six bacterial isolates, identified as *Brevundimonas vesicularis* M4220, *Methylobacterium platani* M7161, *Paracoccus marcusii* M764, *Chryseobacterium soldanellicola* FM524, *Flavobacterium glaciei* MM675, and *Achromobacter insuavis* TS460, were selected to evaluate their antagonistic activity against *A. alternata* in Petri plate antagonism assays. *F. glaciei* strain MM675 demonstrated a strong inhibitory effect with an inhibition rate of 57.64%, while *B. vesicularis* strain M4220, *A. insuavis* strain TS460, and *M. platani* strain M7161 showed moderate inhibitory effects with inhibition rates of 38.54%, 34.03%, and 27.28%, respectively. Additionally, *P. marcusii* strain M764 and *C. soldanellicola* strain FM524 had a weaker inhibitory effect with an inhibition rate of 18.06% and 13.89% (Fig. 4B and Table S10).

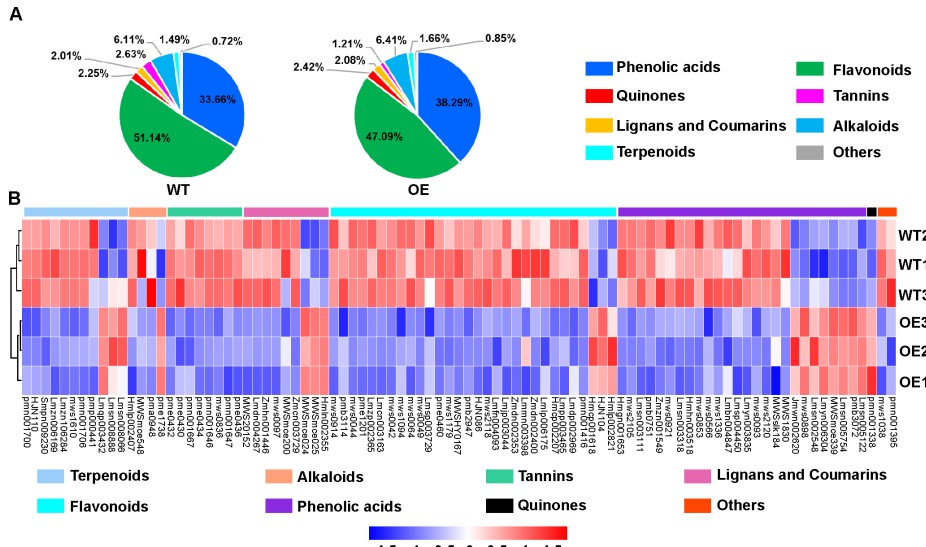

FIG 5 Overall metabolite class composition and differential metabolites identified in *Populus* leaves from *PsnWRKY70*-OE and WT lines. (A) The percentage of the relative content of metabolites in each category accounting for the total amount from the leaves of WT and OE lines of *Populus* inoculated with *A. alternata*. (B) Heatmap of the relative abundance of discriminant metabolites between OE and WT lines of *Populus* inoculated with *A. alternata*.

## Effect of key differential metabolites on the growth of six antagonistic strains

To determine the metabolite variation patterns in the leaves of the WT and OE lines inoculated with *A. alternata*, we employed a widely targeted metabolomics technique and identified the metabolites in the two *Populus* genotypes. A total of 638 metabolites, including 222 phenolic acids, 220 flavonoids, 18 quinones, 50 alkaloids, 13 tannins, 46 terpenoids, 31 lignans and coumarins, and 38 others, were detected and annotated from the leaf samples of the two *Populus* genotypes (Fig. 5A and Table S6). The leaves of WT and OE lines exhibited the higher relative content of phenolic acids, flavonoids, and alkaloids, indicating that the types and distribution of metabolites in the leaves of the two *Populus* genotypes inoculated with *A. alternata* were generally similar (Fig. 5A).

Principal component analysis (PCA) revealed a significant difference in leaf metabolites between the WT and OE lines inoculated with *A. alternata* (Fig. S5). The first two principal components together explained 78.7% of phyllosphere metabolite variance. A distinct separation between the WT and OE lines was noted across the first principal coordinate. The differences in metabolite abundance in the leaves of WT and OE lines inoculated with *A. alternata* were further detected using orthogonal partial least squares discriminant analysis (OPLS-DA). In total, 91 differential metabolites were identified, of which 19 were significantly increased and 72 were significantly decreased in the leaves of the OE line compared to the WT line. Of the differential metabolites, the following showed increased accumulation in the OE line: eight phenolic acids, three flavonoids, three lignans and coumarins, three terpenoids, one alkaloid, and one quinone (Fig. 5B and Table S7).

Among these, the coumarin compounds may play a role in modulating the community composition of the plant microbiome by influencing the proliferation of microbial species (34, 35). Therefore, fraxin (MWSmce025) and scopolin (MWSmce024), which belonged to the coumarin compounds, were selected to further determine their impact on the proliferation of six biocontrol bacterial isolates. The scopolin and fraxin contents were significantly increased in the leaves of the OE line inoculated with *A. alternata* compared to the WT line (Fig. 6A). To determine whether the *PsnWRKY70* gene directly regulated coumarins synthesis, thereby influencing the abundance of microbial taxa in the phyllosphere, a correlation analysis was conducted between the relative abundances

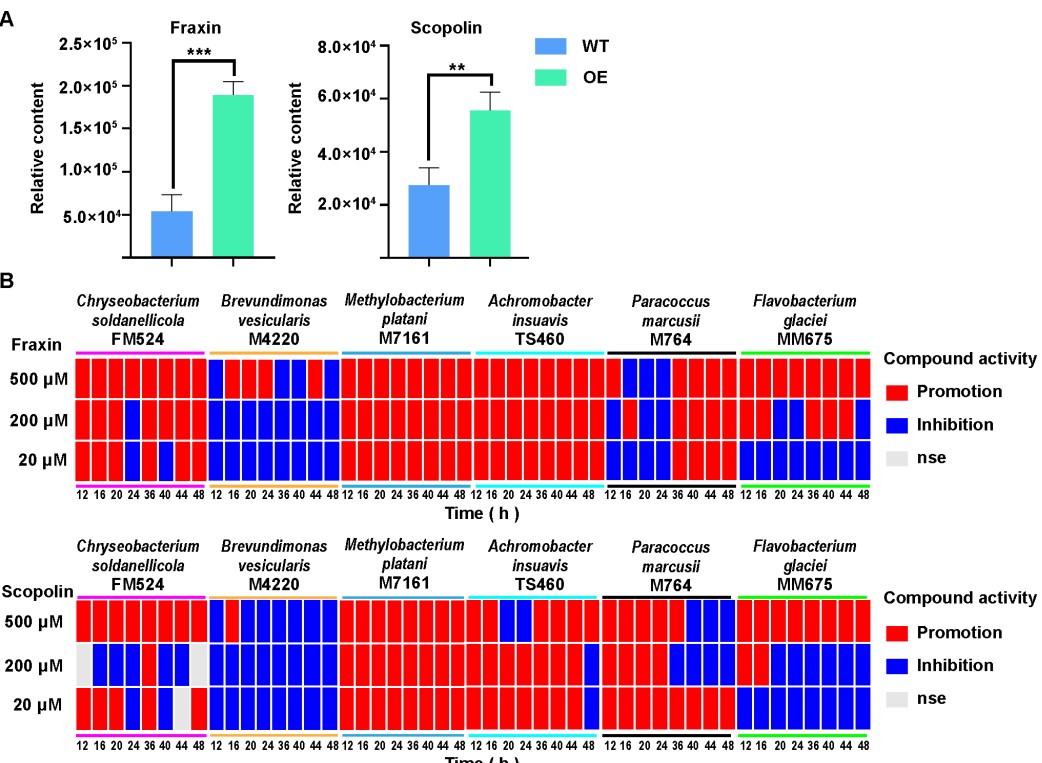

**FIG 6** Fraxin or scopolin content and the effect of fraxin or scopolin on the proliferation of six antagonistic isolates. (A) Fraxin or scopolin content in the leaves of *PsnWRKY70*-OE and WT lines of *Populus*. **$P < 0.01$. (B) Heatmap showing the effect of six antagonistic isolates on differentially accumulated metabolites of the coumarin class (fraxin or scopolin). The red or blue color means that the proliferation of bacterial strains is significantly promoted or inhibited, respectively, in 1/10 Luria-Bertani (LB) medium with fraxin (or scopolin) compared to 1/10 LB medium without fraxin (or scopolin). "nse" indicates that there are no significant differences in the proliferation of bacterial strains cultured in 1/10 LB medium between fraxin (or scopolin) presence or absence.

of the genera from the six aforementioned bacterial isolates with the antagonistic activity against *A. alternata* in the phyllosphere microbial communities and the fraxin and scopolin contents in the leaves of the WT and OE lines inoculated with *A. alternata* (Fig. S6). The relative abundances of these six bacterial genera displayed a significant positive correlation with the fraxin and scopolin contents.

To further validate the impact of coumarins on the six antagonistic bacterial isolates, these bacterial strains were cultured with the compounds fraxin and scopolin. Based on the role of scopoletin concentrations in shaping the assembly of the rhizosphere microbial community (36) as well as the scopoletin and scopolin concentrations accumulated in tobacco (37), the concentrations of 20, 50, and 500 µM were set to analyze the effects of scopolin and fraxin on the proliferation of the six biocontrol strains. Most of the bacterial strains tested exhibited growth promotion in the presence of fraxin or scopolin (Fig. 6B). Both *M. platani* M7161 and *A. insuavis* TS460 used fraxin and scopolin (20 µM, 200 µM, and 500 µM) for the proliferation (Fig. 6B), and these two bacterial genera acted as the connectors in the molecular ecological network of the phyllosphere microbiome of the OE line (Table S5). Fraxin showed a promoting effect on the growth of *C. soldanellicola* FM524, while having no significant effect on the growth of *P. marcusii* M764. However, only the concentration of 500 µM fraxin promoted the growth of *F. glaciei* MM675. Besides the concentration of 200 µM scopolin, scopolin mainly exhibited a promoting effect on the growth of *C. soldanellicola* FM524 and *P. marcusii* M764. Nevertheless, only the concentration of 500 µM scopolin promoted the proliferation of *F. glaciei* MM675. Furthermore, only *B. vesicularis* M4220 was almost

completely inhibited, indicating that the compounds tested could participate in the active selection of phyllosphere bacteria in *Populus* (Fig. 6B). Thus, we concluded that the *PsnWRKY70* gene regulated the biosynthesis of fraxin and scopolin, selected the key taxa, and assembled the phyllosphere microbiome in *Populus*.

## DISCUSSION

The plant genotype can play an important role in regulating certain beneficial phyllosphere microorganisms to defend against pathogen infection (16, 38). In this study, a number of major bacterial genera, including *Pseudomonas*, *Sphingomonas*, *Sphingobium*, *Paracoccus*, and *Pseudonocardia*, constituted the core phyllosphere microbial taxa in OE lines of *Populus* after inoculation with *A. alternata* and could be linked to the inhibition of pathogen growth and reproduction of pathogens on leaf surfaces. For example, *Pseudomonas* spp. can inhibit plant diseases (39, 40), induce systemic host resistance, and produce bioactive molecules to induce stomatal closure, avoiding pathogen entry into the apoplast (41). *Sphingomonas* spp. possess multifaceted functions that protect plants from diseases, increasing the expression of host immune signaling genes, enhancing substrate competition, and producing plant growth-stimulating factors (5, 42, 43). *Pseudonocardia* spp. have been reported to defend against pathogens through the synthesis of antibiotics, limiting pathogen colonization (44). *Paracoccus* spp. and *Sphingobium* spp. also inhibit the growth of the pathogenic fungus *Globisporangium ultimum* and *Pythium* spp., respectively (45, 46). Using four alpha-diversity indices, this study showed that the phyllosphere bacterial community had higher diversity and richness in the OE line than in the WT line after inoculation with *A. alternata*, demonstrating that resistant lines can attract and assemble a relatively complex diverse bacterial community in the phyllosphere to withstand pathogen infection (47). Therefore, infection by *A. alternata* can influence variations in the phyllosphere microbiome community in *Populus*, and plant resistance genotypes have a decisive influence on the structure of the phyllosphere microbiome.

A previous study found that when the relative abundance of *Alternaria* was too high, the relative abundance of beneficial bacteria was unlikely to return to a normal level. The phyllosphere microbiomes were unable to regain homeostasis, leading to leaf damage (48). The co-occurrence network was used to further explain the interactions among complex microbial communities and the relationships between microbial interactions and pathogen infection. In our study, the relationships of phyllosphere bacterial populations within the network from the OE line exhibited higher connectedness, which may contribute to the reaction speed of microbial communities to pathogen disturbances (49). After inoculation with *A. alternata*, both the number of total links and nodes and the topological properties in the co-occurrence network of the phyllosphere microbiomes in the WT line decreased significantly. This could be attributed to the disturbance in the balance of the phyllosphere microbial community by *A. alternata*, which caused the community structure to shift significantly. A previous study has similarly shown that angular leaf spot of cucumber can disrupt the stability of the phyllosphere microbial community network, further affecting phyllosphere functioning (50). However, after inoculation with *A. alternata*, the phyllosphere microbiome in the OE line of *Populus* still had a higher modularity, indicating that it contributes to the rapid response of the microbial community to defend against pathogen infection and reduce the external disturbance (51–53). In addition, there were more connectors in the co-occurrence network of the phyllosphere microbiome in *Populus*, indicating that more key taxa acted to connect each module and build the network with stronger inter-module communication (32, 54). Plants with resistant genotypes frequently have a more complex phyllosphere microbial community network (55). The phyllosphere microbiome of the OE line inoculated with *A. alternata* had more positive associations, indicating that it may be associated with more stable microbial community structure due to the mutualistic relationships within modules (56). These results provide evidence that the *PsnWRKY70* gene in *Populus* participates in plant resistance to pathogen infection by

shaping the microbial community network in the phyllosphere. However, a combination of metabolomics, culturomics, and *in vitro* antagonistic activity assays is needed to further decipher the mechanisms by which the genotype of the host plant shapes the phyllosphere microbiome community.

Plant leaves produce a variety of chemicals that recruit beneficial microbial species and shape phyllosphere microbial communities to potentially enhance their ability to fight pathogens; this is considered the "cry for help" strategy in plants (57, 58). In this study, the metabolites in *Populus* leaves differed between the WT and OE lines after inoculation with *A. alternata*, and 19 metabolites were dominantly enriched in the leaves of the OE line. Among these metabolites, a higher abundance of coumarins (fraxin and scopolin) was detected in the leaves of the OE line after inoculation with *A. alternata*. As signaling molecules, coumarins inhibit a variety of plant pathogens and regulate the interaction between symbiotic microbes, pathogens, and plants (34, 59). They may be "key" components that recruit potentially beneficial phyllosphere microorganisms involved in pathogen resistance. Coumarin biosynthesis originates from the phenyl-propanoid pathway, which facilitates the conversion of phenylalanine to p-coumaroyl-CoA (34). The feruloyl-CoA 6'-hydroxylase1 gene (*F6'H1*), encoding a key enzyme in scopoletin biosynthesis, converts feruloyl CoA to 6'hydroxy-feruloyl CoA. Subsequently, 6-hydroxy-feruloyl CoA undergoes isomerization of the side chain and lactonization to synthesize scopoletin (60, 61). Scopoletin is then converted into scopolin through the catalysis of Uridine diphosphate (UDP)-glucosyltransferase. Scopoletin can also be hydroxylated by scopoletin 8-hydroxylase (*S8H*) to generate fraxetin (62). *NaWRKY70* in *Nicotiana attenuata* is crucial for *A. alternata*-induced production of scopoletin and scopolin by directly binding to and activating the promoter of feruloyl-CoA 6'-hydroxy-lase 1 (*NaF6'H1*), which encodes a key enzyme in scopoletin and scopolin biosynthesis (63). The root-associated microbiome of the scopoletin biosynthesis mutant *f6'h1* of *Arabidopsis thaliana* has demonstrated that scopoletin selectively affects the rhizosphere microbial community assembly (36). A similar result was observed in another study, in which coumarin was found to have a stronger recruitment ability than other secondary metabolites on the phyllosphere microbial community in poplar (64). Thus, we speculated that *PsnWRKY70* was closely related to the assembly of the phyllosphere beneficial microbial community in *Populus* by regulating the synthesis of coumarins (fraxin and scopolin) to resist pathogen infection. Further confirmation is still needed to validate the correlation between coumarins and these differential species of the phyllosphere bacterial community of the OE line.

Based on LEfSe analysis, 29 differential taxa contributed to the structure of the phyllosphere bacterial community of the OE line, suggesting that these bacteria are likely to help the OE line resist *A. alternata* infection. To verify the function of differential taxa in the phyllosphere of the OE line, we isolated culturable bacteria from leaf samples and obtained six bacterial strains annotated to taxa with antagonistic effects on *A. alternata*. Many beneficial functions of these antagonistic bacterial species have been reported, including the secretion of antimicrobial substances, the production of enzymatic activities associated with fungal cell wall degradation (65), the increase in antioxidant enzyme activity of plants (66–68), the triggering of the expression of plant defense genes (69), and plant growth promotion (70). This could potentially serve as a way to help *Populus* resist *A. alternata* infection. Considering the OE line with resistance against *A. alternata* infection and coumarin synthesis in its leaves, coumarins may emerge as potential regulators of the phyllosphere microbiome composition. Most of the six isolates with inhibitory effects against *A. alternata* were significantly enriched in the presence of coumarins. These genera have previously been found to utilize coumarins as a carbon source for growth (71, 72). Other studies have also shown that coumarins can stimulate the growth, chemotaxis, and colonization of probiotic rhizobacteria (36, 73). Thus, we considered that coumarins, as nutrients or proliferating agents, can be potent modulators of the phyllosphere microbiome assembly to enhance the niche competition of beneficial microbial species. Interestingly, the genera *Methylobacterium*

and *Achromobacter*, acting as pathogen antagonists, were also identified as connectors in the phyllosphere microbiome of the OE line using network analysis, suggesting an important role in network structure (74). Based on results of the phyllosphere microbiome sequencing, metabolomics, and plate antagonistic assays, we concluded that these phyllosphere metabolites, acting as nutrients or proliferating agents, selectively regulate the growth and colonization of potential key species, thereby modifying the phyllosphere microbiome and preventing infection by *A. alternata*. Although the role of coumarins in the ability of the *Populus* phyllosphere microbiome assembly to resist *A. alternata* was preliminarily demonstrated, the defense mechanisms of the other differential metabolites in leaves of the OE line remain unknown because of the lack of comprehensive research. Further research on the relationships between the role of specific metabolic compounds in leaves regulated by the *PsnWRKY70* gene and phyllosphere beneficial microbiomes should be performed to verify the effect of the plant genotype on these "potentially important" bacterial species for protecting *Populus* from *A. alternata* infection.

## Conclusions

In this study, we investigated the microbial community composition and metabolic profiles of *Populus* leaves after inoculation with *A. alternata*, revealing the potential defense strategies of resistant plant genotypes. The invasion of *A. alternata* distinctly altered the diversity and composition of the phyllosphere microbiome in *Populus*, the stability and complexity of the molecular ecological network, and the number of connector species. The OE line shaped the phyllosphere microbial community assembly to resist infection by *A. alternata* by adjusting metabolite synthesis in the leaves. Among the differential metabolites, coumarins (e.g., fraxin and scopolin) acted as nutrients or proliferating agents for key biocontrol bacterial species and affected important phyllosphere connector species in the molecular ecological network. Together, understanding how the *PsnWRKY70* gene shapes the phyllosphere microbial community in *Populus* expands our understanding of disease resistance mechanisms in phyllosphere bacterial communities and could provide a basis for disease resistance breeding in forests. In the future, more differential metabolites from the OE line should be investigated to determine the comprehensive recruitment mechanisms in the phyllosphere microbiome, enabling the plant to resist *A. alternata* infection.

## MATERIALS AND METHODS

### Plant sampling

The WT and OE lines of 3-year-old field poplar (*P. simonii* × *P. nigra*) located in the Breeding Base of Northeast Forestry University in Harbin, Heilongjiang Province (126°640 E, 45°720 N), were used as experimental materials. The transgenic *PsnWRKY70*-OE lines of poplar used in this study were previously described (75). The open reading frame of *PsnWRKY70* was amplified from the cDNA of *P. simonii* × *P. nigra* based on the sequence information of *Populus trichocarpa* (Potri.016 G137900). The CaMV 35S promoter-derived pGWB5-*PsnWRKY70*-GFP OE vector was constructed. OE vectors were transferred into *P. simonii* × *P. nigra* to obtain OE lines using the *Agrobacterium tumefaciens*-mediated leaf-disc transformation method (75). One-year-old branches of the WT and OE lines were collected outside and brought back indoors on 16 May 2021. These branches were cut into 15 cm-long brachyblasts and then inoculated with *A. alternata* after 30 days of growth in the greenhouse (16/8 h light/dark, 25°C/20°C, and 50%–60% relative humidity) (24). The expression levels of the *PsnWRKY70* gene were determined in leaf samples of the WT and OE lines, both non-inoculated and inoculated with *A. alternata* (0, 6, 12, 24, and 36 h) via qRT-PCR. Total RNA of each sample was extracted using a Universal Plant Total RNA Extraction Kit (BioTeke Corporation, Beijing, China), and cDNA was separately synthesized using the Toyobo Reverse Transcription Kit (ReverTra AceqPCR RT Master Mix

with gDNA Remover; Toyobo, Osaka, Japan). qRT-PCR was performed on an ABI-7500 quantitative PCR instrument using Toyobo SYBR Green Real-time PCR Master Mix Plus (Toyobo, Osaka, Japan) (75). 18S was used as the internal control to normalize the expression levels of the *PsnWRKY70* gene. The relative expression levels of the target gene were calculated using the $2^{-\Delta\Delta CT}$ method as described previously (75). *A. alternata* preserved in the laboratory was used to inoculate potato dextrose agar (PDA) plates and cultured in the dark at 25°C for 15 days for spore collection (24). Conidia were collected by adding 3 mL of sterile distilled water to each plate and gently brushing the colonies. The colonies were gently brushed, and the suspension was filtered through two layers of sterile gauze to remove mycelial fragments. This procedure was repeated twice (76). To evaluate the disease resistance of poplar plants, a spore suspension of pathogenic *A. alternata* with $5.0 \times 10^6$ spores/mL was used to inoculate the leaves of the healthy OE and WT lines using the spray method, and the OE and WT plants sprayed with water were used as controls (76). Leaf samples of the WT and OE plants that were not inoculated and inoculated with *A. alternata* were collected 15 days after inoculation, placed in 50 mL sterile plastic tubes, and quickly transported to the laboratory for phyllosphere bacterial community and leaf metabolomics analyses. Leaf samples (five leaves per plant) of each treatment were collected from 10 plants and used to analyze phyllosphere bacterial communities, except for the OE plants inoculated with *A. alternata*, for which samples were only collected from nine plants. Leaf samples from the WT and OE lines inoculated with *A. alternat*a were collected for widely targeted metabolomics analysis. Each biological replicate sample was composed of 15 leaves pooled from three plants, and three independent biological replicates were performed.

## DNA extraction, PCR amplification, and sequencing

To evaluate the phyllosphere bacterial community, the surfaces of *Populus* leaf samples were wiped repeatedly with sterile cotton swabs containing 0.1% Tween 20 (77). Cotton swabs containing the phyllosphere bacterial community from the same tree were mixed into one sample. Swabs were used to extract genomic DNA of the phyllosphere microbial community using a DNeasy Power Soil Kit (QIAGEN, Hilden, Germany) following the manufacturer's instructions.

The V4 region of the bacterial 16S rRNA gene was amplified with the barcoded primer set 515F (5′-GTGCCAGCMGCCGCGGTAA-3′) and 806R (5′- GGACTACHVGGGTWTC-TAAT-3′) in a 25 mL reaction system, containing 1.0 µL of genomic DNA, 12 µL of PCR water (QIAGEN, Hilden, Germany), 10 µL of 5Prime Hot Master Mix (Quantabio, Beverly, Massachusetts, USA), and 1.0 µL of each forward/reverse primer (10 nM final concentration). The PCR program was run as follows: 94°C for 3 min; 35 cycles of 94°C for 45 s, 50°C for 60 s, and 72°C for 90 s; and final extension at 72°C for 10 min. Three PCR amplicon products from the same sample were mixed in equal proportions and quantitatively analyzed using a Quant-iT dsDNA Assay Kit (Invitrogen, Eugene, Oregon, USA). Mixed PCR products were purified and then sequenced on an Illumina MiSeq platform by Shanghai Personal Biotechnology Co., Ltd (Shanghai, China).

## Sequence data analysis

The raw sequencing data obtained were processed on the QIIME2 (2020.2) platform (78). Sequence quality control, including de-noising, trimming, and chimera removal, was performed using DADA2 (79). ASVs were clustered based on a 99% similarity cutoff using the QIIME2 feature classifier (80).

## Isolation and molecular identification of poplar phyllosphere bacteria

Healthy *Poplar* leaves were collected and cut into pieces with sterile scissors and then ground into a homogenate in a mortar with 2 mL of sterile water. Bacteria were isolated and purified on MYX, R2A, MM + MeOH, Flour, TSB, YEM, TYG, M408, M715, and TWYE agar media after serial dilutions (81) and incubated in the dark at 30°C. Each bacterial

isolate was cryopreserved at −80°C in a bacterial suspension with 60% glycerol (1:1, [vol/vol]).

A single colony of each bacterial isolate was cultured overnight at 30°C with shaking. Genomic DNA was extracted from collected bacterial cells using a Bacterial Genomic DNA Extraction Kit (Tiangen, DP302-02, Beijing, China) for molecular identification with the PCR technique. The 16S rRNA gene was amplified in a 25 µL PCR reaction using the universal primers 27F (5′ AGAGTTTGATCATGGCTCAG-3′) and 1492R (5′ TACGGTTACCTTGT TACGACTT-3′). The PCR mixture contained 1 µL of template DNA, 12.5 µL of master mix, 1 µL of each of the forward and reverse primers, and 9.5 µL of sterile deionized water. The bacterial 16S rRNA gene was amplified using the following PCR program: 95°C for 5 min; 30 cycles of 95°C for 30 s, 55°C for 30 s, and 72°C for 1 min; and final extension at 72°C for 10 min (82). The PCR products were sequenced using the 27F primer at Tsingke Biotechnology Co., Ltd (Beijing, China). The obtained sequences were analyzed using the Basic Local Alignment Search Tool on the National Center for Biotechnology Information website (https://blast.ncbi.nlm.nih.gov/Blast.cgi) and by Sequence Match on the Ribosomal Database Project website (http://rdp.cme.msu.edu).

### *In vitro* antagonistic bioassays against *A. alternata*

The *in vitro* antifungal activity of bacterial isolates against *A. alternata* was screened on PDA medium plates. A mycelial disk with a diameter of 2 mm was cut from the edge of *A. alternata* cultured for 7 days and placed 1 cm from the edge of a new PDA plate ($\varphi$ = 90 mm) (83). Each isolated bacterial suspension was streaked across the middle of the plate, and three replicate plates were prepared. Only mycelial disks were used as the control. The plates were incubated in the dark at 25°C for 7 days, and the radius of the fungal colony was recorded. The fungal inhibition rate of *A. alternata* was calculated as follows:

$$\text{Fungal inhibition rate (\%)} = \frac{\text{Colony diameter of control-Colony diameter of treatment}}{\text{Colony diameter of control}} \times 100$$

### Widely targeted metabolomics and data processing

Forty-five days after inoculation with an *A. alternata* spore suspension ($10^5$ cells per mL), leaves of OE and WT poplar lines were collected, wrapped in aluminum foil, quickly frozen in liquid nitrogen, and stored at −80°C until metabolite extraction. After low-temperature freeze-drying treatment with a vacuum freeze dryer (Shanghai, China, Scientz-100F), the collected leaf samples were lyophilized and pulverized at 30 Hz for 1.5 min using a mixing mill with zirconia beads (Shanghai, China, MM 400, Retsch). A 100 mg sample of the lyophilized powder was dissolved in 1.2 mL of 70% methanol solution, vortexed six times for 30 s every 30 min, and placed in a refrigerator at 4°C overnight. After centrifugation at 12,000 rpm for 10 min, the extracts were filtered (SCAA-104, 0.22 µm pore size; ANPEL, Shanghai, China) and analyzed using the Ultra-performance liquid chromatography-tandem mass spectrometry (UPLC-MS/MS) system (UPLC, Shanghai, China, SHIMADZU Nexera X2; MS, Shanghai, China, Applied Biosystems 4500 Q TRAP) equipped with a C18 column (Shanghai, China, Agilent SB-C18, 1.8 µm, 2.1 × 100 mm). Mobile phase A was pure water with 0.1% formic acid, and mobile phase B was acetonitrile with 0.1% formic acid.

The measurement of the extracts from poplar leaf samples was performed as previously described (84). Linear ion trap and triple quadrupole scans were acquired on a triple quadrupole-linear ion trap mass spectrometer, AB4500 Q TRAP UPLC/MS/MS System, equipped with an ESI Turbo Ion-Spray interface, and controlled using Analyst 1.6.3 software (AB Sciex, Framingham, MA, USA). The ESI source operation parameters were based on the method described by Wang et al. (85). UPLC-MS/MS baseline data analysis and screening of significantly changed metabolites followed the methods described by Li et al. (86).

Multivariate statistical analysis was used to perform PCA. PCA was performed using the statistics function prcomp in R (www.r-project.org). The data were unit

variance scaled before unsupervised PCA. The stability and reliability of the model were predicted using OPLS-DA. Significantly regulated metabolites between groups were determined using the following threshold: variable importance in projection (VIP) ≥1 and absolute $\log_2FC$ (fold change) ≥1. VIP values were extracted from the OPLS-DA results (87). Identified metabolites were annotated using the Kyoto Encyclopedia of Genes and Genomes (KEGG) Compound Database (http://www.kegg.jp/kegg/compound/), and annotated metabolites were mapped to the KEGG Pathway Database (http://www.kegg.jp/kegg/pathway.html).

## Effects of scopolin and fraxin on the growth of six antagonistic bacterial strains

The scopolin and fraxin contents in the leaves of the WT and OE plants were detected based on UPLC-MS/MS analyses, and the peak intensity values were normalized using ln transformation (88).

Scopolin and fraxin (Shanghai MacLean Biochemical Technology Co., Ltd.) were prepared in different concentrations (25, 10, and 1 mM diluted in dimethylsulfoxide) to evaluate the effect on the growth of six antagonistic bacterial strains. A volume of 196.0 µL of six bacterial suspensions with absorbance values at 600 nm ($OD_{600}$) of 0.01 (diluted with 1/10 LB medium) was pipetted into a 96-well plate, and 4.0 µL of fraxin and scopolin solutions (1, 10, and 25 mM) were added to obtain final concentrations of 20, 200, and 500 µM, respectively. The plates were then incubated at 30°C. The growth of six antagonistic bacteria was determined by measuring the $OD_{600}$ after 0, 12, 16, 20, 24, 36, 40, 44, and 48 h using a Synergy 96-well plate reader (BioTek, Winooski, Vermont, USA) (89, 90). Six biological replicates were prepared for each treatment.

## Statistical analysis

Four alpha-diversity indices of the phyllosphere bacterial microbial community were calculated using the "vegan" packages in R (v4.3.1) and were visualized in boxplots by using the "ggplot2" and "ggpubr" packages in R (v4.3.1). PCoA based on Bray–Curtis dissimilarity was performed using the "vegan" package in R (v4.3.1) (91). To evaluate the variance between different treatment groups, PERMANOVA was performed using the "adonis" function with 999 permutations. LEfSe (http://huttenhower.sph.harvard.edu/galaxy/) was used to identify the biomarker taxa for different groups, using linear discriminant analysis threshold score of 2.0, with statistically significant differences (92). Co-occurrence network analysis was performed using the Molecular Ecology Network Analysis Pipeline (http://ieg4.rccc.ou.edu/MENA/) under RMT. The networks were visualized using Gephi software (v0.9.2). According to the within-module connectivity ($Zi$) and among-module connectivity ($Pi$), the topological roles of nodes were divided into four categories: network hubs ($Zi > 2.5$, $Pi > 0.62$), module hubs ($Zi > 2.5$, $Pi \leq 0.62$), connectors ($Pi > 0.62$, $Zi \leq 2.5$), and peripherals ($Zi \leq 2.5$, $Pi \leq 0.62$) (53). Student's $t$-test was used for statistical comparisons of four alpha-diversity indices and the relative abundances of the dominant species among treatments at the phylum and genus levels. The correlation was calculated and visualized using the "ggplot2," "pppubr," and "ggmisc" packages, and Spearman rank correlation was used to gauge the relationships between the relative fraxin or scopolin contents and the relative abundances of the genera from the six biocontrol bacteria isolates. Significant differences at the phyla or genera level were analyzed using two-sided Welch's $t$-test with a $P$-value of 0.05 in STAMP software (v2.1.3).

## ACKNOWLEDGMENTS

We thank members of the G.L. and B.N. laboratories for valuable advice.

This research was funded by STI 2030-Major Projects (2023ZD0405603; to G.L.) and Key R&D Plan Projects in Xinjiang Uygur Autonomous Region (grant number 2022B02014; to B.N. and D.W.).

W.W., W.W., J.J., B.N., and G.L. designed the experiments. X.B., K.C., X.Z., and W.W. contributed the materials. W.W., K.C., X.Z., and X.B. performed the experiments. W.W., W.W., J.Y., D.W., and B.N. analyzed the data and wrote the draft manuscript. W.W., D.W., and B.N. revised and polished the manuscript. All authors contributed to the article and approved the submitted version.

## AUTHOR AFFILIATIONS

[1]State Key Laboratory of Tree Genetics and Breeding, Northeast Forestry University, Harbin, China
[2]Peking University Institute of Advanced Agricultural Sciences, Shandong Laboratory of Advanced Agriculture Sciences in Weifang, Weifang, Shandong, China
[3]College of Life Sciences, Shandong Agricultural University, Tai'an, Shandong, China
[4]College of Life Science, Northeast Forestry University, Harbin, China
[5]The Center for Basic Forestry Research, Northeast Forestry University, Harbin, China
[6]State Key Laboratory of Plant Cell and Chromosome Engineering, Institute of Genetics and Developmental Biology, Innovation Academy for Seed Design, Chinese Academy of Sciences, Beijing, China
[7]School of Life Sciences, Qilu Normal University, Jinan, China

## AUTHOR ORCIDs

Wei Wang http://orcid.org/0000-0003-0510-4872
Weixiong Wang http://orcid.org/0009-0002-5157-6846
Jing Jiang http://orcid.org/0000-0001-7845-3307
Xiangdong Bai http://orcid.org/0009-0009-3158-8541
Kun Chen http://orcid.org/0000-0001-9291-9732
Xiaoyue Zhang http://orcid.org/0009-0002-8404-9794
Jingya Yang http://orcid.org/0009-0007-1688-6407
Di Wu http://orcid.org/0000-0002-1310-8921
Ben Niu http://orcid.org/0009-0008-6567-6833
Guifeng Liu http://orcid.org/0000-0002-5468-6293

## FUNDING

| Funder | Grant(s) | Author(s) |
|---|---|---|
| STI 2030-Major Projects | 2023ZD0405603 | Guifeng Liu |
| Key R&D Plan Projects in Xinjiang Uygur Autonomous Region | 2022B02014 | Di Wu |
|  |  | Ben Niu |

## DATA AVAILABILITY

The original contributions presented in the study are included in the article/Supplementary Material. Further inquiries can be directed to the corresponding author. The sequencing data of the 16S rRNA gene survey reported in this paper have been deposited in the NCBI Sequence Read Archive (BioProject ID no. PRJNA1158217, accession nos. SAMN 43532619-SAMN 43532657). GenBank accession number(s) for six antagonistic strains nucleotide sequence(s): PQ376604-PQ376609.

## ADDITIONAL FILES

The following material is available online.

### Supplemental Material

**Supplemental Figures (mSystems01765-24 s0001.docx).** Figures S1 to S6.
**Supplemental Tables (mSystems01765-24 s0002.xlsx).** Table S1 to S12.

Open Peer Review

**PEER REVIEW HISTORY (review-history.pdf).** An accounting of the reviewer comments and feedback.

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
