## [Reviewer comments · mSystems]

Populus simonii* × *P. nigra* overexpressing *PsnWRKY70* recruits phyllosphere bacterial strains that inhibit *Alternaria alternata

Wei Wang, Weixiong Wang, Jing Jiang, Xiangdong Bai, Kun Chen, Xiaoyue Zhang, Jingya Yang, Di Wu, Ben Niu, and Guifeng Liu

Corresponding Author(s): Ben Niu, Northeast Forestry University

Review Timeline:

Submission Date:	December 31, 2024
Editorial Decision:	February 25, 2025
Revision Received:	May 27, 2025
Editorial Decision:	July 1, 2025
Revision Received:	July 13, 2025
Accepted:	July 22, 2025

Editor: Juliana Almario

Reviewer(s): The reviewers have opted to remain anonymous.

Transaction Report:

DOI: <https://doi.org/10.1128/msystems.01765-24>

Re: mSystems01765-24 (*Populus simonii* × *P. nigra* overexpressing *PsnWRKY70* recruits phyllosphere bacterial strains that inhibit *Alternaria alternata*)

Dear Dr. Ben Niu:

Thank you for sending your work to mSystems. Below you will find my comments, instructions from the mSystems editorial office, and the reviewer comments.

This manuscript has potential for publication at mSystems, but still requires additional information. We appreciate how the study uses multiple approaches to characterize the microbial communities associated with poplar trees, and how transgenics and pathogen inoculation changes them. Furthermore, you identify some potential metabolites involved in the interaction. Three experts have reviewed this manuscript, and all their concerns would need to be addressed before publication. In particular, the reviewers noted that the choice of fraxin and scopolin are not explained, and other important details are missing as well, including how specific bacteria were selected for inhibition tests and other necessary methods.

Beyond the reviewer concerns, I see that there is no Data Availability paragraph at the end of the Methods section. We require that you comply with ASMs Data Policy in order for a manuscript to be published. Lastly, I noticed that two very similar papers were recently published by your team, and are not cited here (<https://doi.org/10.1016/j.pmpp.2024.102349> and <https://doi.org/10.1016/j.pmpp.2024.102461>). Please include them as citations and outline how this manuscript expands on that research in your response letter.

Revision Guidelines

Sincerely,

Reviewer #1 (Comments for the Author):

The paper entitled "Populus simonii × P. nigra overexpressing PsnWRKY70 recruits phyllosphere bacterial strains that inhibit *Alternaria alternata*" was carefully reviewed. In this work, the authors demonstrated that PsnWRKY70 gene involved in the plant immunity in the recruitment of specific phyllosphere microbiome against the fungal pathogen in *Populus simonii* × *P. nigra*. The authors attempted to reveal the mechanisms underlying this process by using the 16S rRNA gene amplicon sequencing combined with the plant metabolomics analysis and culturomics techniques.

Here are some concerns that need to be addressed:

- 1.Lines 35-40: these two sentences "To address this...that it had stronger robustness to withstand pathogen infection" are too complex to understand. It is suggested that they be rewritten and combined.
- 2.Lines 120-124: the three objectives proposed in the "Introduction" section, objectives ii) and iii) are somehow overlapped. It is suggested that they be rewritten and think about a new objective iii) here.
- 3.Line 194: what is RMT short for? please provide its full name in the text.
- 4.Lines 239-240: Here, 568 phyllosphere bacterial strains were isolated. please provide more information about these phyllosphere bacterial strains in the text or Supplementary Materials , such as, How much did they cover the taxa listed in the sequencing data?
- 5.Lines 263-275: it is suggested that "the 91 differential metabolites" should be listed in the supplementary materials, with the amounts of these compounds in the leaves if them available. Addition, why only choose the two coumarin compounds, scopolin and fraxin, for the bacterial growth test?
- 6.Lines 289-291:there were not experimental evidences for the regulation of the biosynthesis of fraxin and scopolin in the leaves by PsnWRKY70. It would be great if the authors will be able to talk about the possible mechanisms of this process in the "Discussion" section.
- 7.Line 294: the "Discussion" section contains too much redundant information. The text should be reduced by focusing on the novel contribution of this study. The authors might consider talking about the potential pathway of the modulation of the biosynthesis of the two coumarin compounds and the mechanisms underlying the responses of the antagonistic bacterial strains to these chemicals. In addition, the authors may consider to cite some recent publications to support their ideas in the Discussion.
- 8.The "Results" section of this manuscript has not been described sufficiently. In particular, in "Effect of key differential metabolites on the growth of six antagonistic strains" (Line 251). In addition, Is there a clear correlation between the contents of the two compounds and the relative abundances of the ASVs representing these biocontrol strains? Please provide more details.
- 9.Figure 6, why did the authors use 20 µM, 50 µM and 500 µM of the two coumarin compounds for the bacterial growth test? The authors may think about providing rationale behind the decision.

Reviewer #2 (Comments for the Author):

This manuscript aimed to reveal how the metabolites regulated by the PsnWRKY70 gene of *Populus* triggered the phyllosphere microbiome to defense the infection of *Alternaria alternata* by using the multi-omics technologies. The authors found that the phyllosphere microbiome of the wild-type and PsnWRKY70-overexpressing lines showed significant difference when being challenged by *A. alternata*, and that, specifically, the stability and complexity of the bacterial communities was influenced by *A. alternata*. The authors also found that the growth of two isolates belonging to the genera *Methylobacterium* and *Achromobacter*, respectively, was significantly promoted by the coumarin compounds, fraxin and scopolin enriched in the OE line, and that both strains were able to inhibit the growth of *A. alternata*. The results shown in this manuscript are interesting, which expand the knowledge of the functionality of gene PsnWRKY70 in resistance to phytopathogens. Detailed comments are listed as follows:

Major points:

1. L280 to 281: The authors identified quite a few chemicals accumulated in the OE line, why were only farxin and scopolin used to culture the antagonistic strains? Please give the reason(s) for choosing these two compounds in the text.
2. L133 to L171: It is interesting that the compositions and diversities of the phyllosphere bacterial communities was influenced by the inoculation of *A. alternata*. It would be more informative if the authors add some data of the comparison of the phyllosphere microbiota of the healthy plants and the ones inoculated with *A. alternata* regardless of the plant genotypes in the first two parts of the RESULTS and update the Figures 1 and 2.
3. The DISCUSSION of this manuscript is definitely too long. The text of the network analysis will need to be reduced. How *A. alternata* infection influences the phyllosphere microbiome and the mechanisms of the suppressive effects of the six biocontrol strains against *A. alternata* are good points to discuss.

Minor points:

1. L444 to L446: Please provide some parameters of the growth conditions in the greenhouse, for example, temperature, humidity, light and etc.

2. L459 to L462: Please note that the number of the samples for the metabolome analysis is inconsistent with that listed in the supplementary material.
3. L544 to L545: Please provide the results of the PCA analysis on the metabolites in the leaves of OE and WT lines of poplar.
4. L281 and L427: Is it scopoline or scopolin? Please unitize the spelling of this word through the text.
5. L680 to L685: Please indicate the meaning of the asterisks shown in Figures S2 and S3 in their legends.
6. Please clarify the criteria for dividing the topological roles of nodes in MATERIAL AND METHODS, according to the within module connectivity (Z_i) and among module connectivity (P_i).
7. L231: Some data, for example, the inhibition rate of the six antagonistic bacterial strains against *A. alternata* and/or the radii of the colonies of *A. alternata* co-cultured with these biocontrol strains should be described in the text.
8. L700: In Figure 2B, the values of R^2 should be given.
9. It would be great if the authors will ask an English native speaker to improve the language.

Reviewer #3 (Comments for the Author):

The authors investigate the role of PsnWRKY70 gene overexpression in hybrid *Populus* strains in response to *Alternaria alternata* fungus infection, focusing on its impact on phyllosphere microbiome assembly and leaf metabolite composition.

To explore this, they compare the phyllosphere microbiomes of overexpression (OE) and wild-type (WT) *Populus* strains before and after pathogen inoculation. They identify bacterial taxa associated with pathogen growth inhibition and determine which metabolites promote these beneficial bacteria. Their findings reveal that the phyllosphere microbiome of OE strains differs from WT strains post-inoculation and using experiments find several bacterial strains exhibiting pathogen-inhibiting properties. Additionally, they confirm that two metabolites, fraxin and scopoline, enhance the growth of these protective bacteria.

Although the exact molecular link between PsnWRKY70 overexpression and production of specific metabolites is not clear, this study nicely integrates bacterial amplicon sequencing, metabolomics, and experimental validation to highlight and confirm the importance of some potential players in pathogen resistance in OE *Populus* strains. Some technical details in the manuscript need clarification, but overall, I find it a valuable contribution to the literature on fungal resistance mechanisms.

General comments

One of my comments is that authors should explain better how OE poplar was constructed, and how the gene is overexpressed. For example, it is not clear if the gene is overexpressed continuously and by how much compared to WT, and if there are any other molecular differences between OE and WT strains. Do authors know of any other genes that are overexpressed? Authors should specify these details to make sure that gene overexpression is the only factor driving differences in OE strain. Authors should also show or cite previous results demonstrating that gene expression differs between WT and OE, after fungal inoculation.

I found it hard to follow the section on how specific bacteria were selected for experimental inhibition tests. Some important details are missing, for example, authors refer to a phylogenetic method to subset bacterial strains enriched in OE after fungus inoculation, but the analysis is not described anywhere in the methods, and furthermore it's not written how many strains were found to be enriched, and how many strains were selected for the inhibition test. Please provide all necessary details.

Minor comments:

Page2, Abstract - Abbreviations and names are written for the first time without explanation. Please explain what is OE, and what are fraxin and scopoline.

Page 7, lines 145-148: this statement should be supported by the tests, eg those shown in Fig. S2

Fig. S2 - Please add what asterisks mean.

Page 8, lines 163-165: I find this sentence hard to follow. What authors mean by: "the genotype distribution characteristics of host plants after inoculation with *A. alternata*" - please rewrite or clarify.

Page 8, line 179-181: A suggestion - it looks to me that when comparing inoculated with non-inoculated, diversity decreases after inoculation, but to a smaller degree in OE strains. Authors could compare non-inoculated and inoculated strains to test it.

Page 9, line 85 - Authors could add R^2 to show that there is indeed a stronger distinction.

Page 9, line 195-8: "All empirical networks.." Please add a reference to a figure for this sentence.

Page 9, line 203 - "bacterial network of the OE line" - Please specify here if you mean inoculated or non-inoculated one

Page 10 - lines 209-211 - Are you comparing here networks before and after inoculation, or WT and OE networks? From the table it looks like before inoculation both WT and OE networks are most complex.

Page 11 - line 241-243 - Referring to general comments: Please explain how this analysis was done. Also not clear, how many strains were selected for inhibition test.

Figure 5A - Please specify what the scale represents.

Supplementary Table 6 and lines 258-261 - The table does not show which metabolites are enriched. Perhaps you should add a column with enrichment or refer to another figure/result.

Comments and Suggestions for Authors

The paper entitled “*Populus simonii* × *P. nigra* overexpressing PsnWRKY70 recruits phyllosphere bacterial strains that inhibit *Alternaria alternata*” was carefully reviewed. In this work, the authors demonstrated that *PsnWRKY70* gene involved in the plant immunity in the recruitment of specific phyllosphere microbiome against the fungal pathogen in *Populus simonii* × *P. nigra*. The authors attempted to reveal the mechanisms underlying this process by using the 16S rRNA gene amplicon sequencing combined with the plant metabolomics analysis and culturomics techniques. Here are some concerns need to be addressed:

1. Lines 35-40: these two sentences “To address this...that it had stronger robustness to withstand pathogen infection” are too complex to understand. It is suggested that they be rewritten and combined.
2. Lines 120-124: the three objectives proposed in the “Introduction” section, objectives ii) and iii) are somehow overlapped. It is suggested that they be rewritten and think about a new objective iii) here.
3. Line 194: what is RMT short for? please provide its full name in the text.
4. Lines 239-240: Here, 568 phyllosphere bacterial strains were isolated. please provide more information about these phyllosphere bacterial strains in the text or Supplementary Materials , such as, How much did they cover the taxa listed in the sequencing data?
5. Lines 263-275: it is suggested that “the 91 differential metabolites” should be listed in the supplementary materials, with the amounts of these compounds in the leaves if them available. Addition, why only choose the two coumarin compounds, scopolin and fraxin, for the bacterial growth test?
6. Lines 289-291:there were not experimental evidences for the regulation of the biosynthesis of fraxin and scopolin in the leaves by PsnWRKY70. It would be great if the authors will be able to talk about the possible mechanisms of this process in the “Discussion” section.
7. Line 294: the “Discussion” section contains too much redundant information. The

text should be reduced by focusing on the novel contribution of this study. The authors might consider talking about the potential pathway of the modulation of the biosynthesis of the two coumarin compounds and the mechanisms underlying the responses of the antagonistic bacterial strains to these chemicals. In addition, the authors may consider to cite some recent publications to support their ideas in the Discussion.

8. The “Results” section of this manuscript has not been described sufficiently. In particular, in “Effect of key differential metabolites on the growth of six antagonistic strains” (Line 251). In addition, Is there a clear correlation between the contents of the two compounds and the relative abundances of the ASVs representing these biocontrol strains? Please provide more details.
9. Figure 6, why did the authors use 20 μM , 50 μM and 500 μM of the two coumarin compounds for the bacterial growth test? The authors may think about providing rationale behind the decision.

Dear Editor and Reviewer,

We thank you for the time and efforts in evaluating our manuscript. We very much appreciate the comments and suggestions provided and have modified the manuscript accordingly. Before providing a detailed point-by-point response we simply summarize the major changes contained in the new version of the manuscript. As you will see we have carefully revised the entire manuscript. The major changes in the revised version can be summarized as follows:

- 1) To enhance the cohesion of the "ABSTRACT" section, we rewrote the sentences that were difficult to understand. In the "INTRODUCTION" section, we outlined and cited the two papers published by our team, and re - summarized objectives (ii) and (iii).
- 2) To concise the DISCUSSION and make the wording more accurate, we have deleted the part of the discussion that have no meaning or common knowledge points. We have also supplemented the exploration for the potential pathway of the biosynthesis of the scopolin and fraxin, as well as the antibacterial mechanisms of the six biocontrol strains against *A. alternata*.
- 3) To excavate more important and effective information on the RESULTS, we have added elaborate messages including the correlation between the contents of the two compounds and the relative abundances of the ASVs representing these biocontrol strains, the inhibition rate of the six antagonistic bacterial strains against *A. alternata*, and the PCA analysis on the metabolites in the leaves of the OE and WT lines of poplar. At the same time, we have also explained why only farxin and scopolin were used to culture the antagonistic strains and why 20 μM , 50 μM , and 500 μM of these two compounds were used for the bacterial growth test.
- 4) We have provided a more informative description, such as some of the parameters related to plant growth conditions in the greenhouse, the criteria for dividing the topological roles of nodes and the values of R^2 .

In the following pages, we provide a point-by-point response to each of the editor's comments. The original comments are presented in blue font and our responses are in black font and we used red font where we place in the letter the modified text as it appears in the revised manuscript. We have also directly replied to all the comments marked by editor in the text file of the revised manuscript. Beside this response letter, we also attach two comparison files showing the changes we have made on the previous version of text to get the current version of the manuscript. We feel the manuscript has been greatly improved thanks to the thoughtful comments and suggestions you

have provided. We hope you will now find our work suitable for publication in mSystems.

Sincerely,
Wei Wang, Weixiong Wang, Di Wu and Ben Niu (for all authors)

Editor (Comments for the Author):

I noticed that two very similar papers were recently published by your team, and are not cited here (<https://doi.org/10.1016/j.pmpp.2024.102349> and <https://doi.org/10.1016/j.pmpp.2024.102461>). Please include them as citations and outline how this manuscript expands on that research in your response letter.

We thank editor for this rigorous advice. In the article “Assembly of phyllosphere bacterial community with *PsnWRKY70* in poplar”, it has been confirmed that significant differences exist in the bacterial community structure and β - diversity among transgenic *PsnWRKY70* gene over - expression (OE), *PsnWRKY70* gene suppression expression (RE), and wild - type (WT) lines by inoculating a synthetic bacterial community composed of 39 bacterial strains isolated from the phyllosphere of poplar under tissue - culture conditions. In the article “Effects of transcription factor *PsnWRKY70* on phyllosphere bacterial community of *Populus* infected by *Melampsora laricis - populina*”, it has also been confirmed that the *PsnWRKY70* gene can recruit strains, enhancing the ability of the OE lines of transgenic poplar to resist the invasion of *Melampsora laricis - populina*. Although the *PsnWRKY70* gene is known to influence the phyllosphere microbiome, how WRKY70 transcription factors (TFs) regulate the assembly of the microbial community in the phyllosphere to withstand poplar leaf blight caused by *Alternaria alternata* remains unknown.

These two articles have been cited in the revised manuscript through the following text modifications:

Lines 102 to 109: “In previous research, we found that the transcription factor *PsnWRKY70* was involved in regulating the composition of the poplar phyllosphere microbiome. It recruited beneficial microbes to enhance poplar's resistance to *Melampsora laricis – populina* (27). We also found that the *PsnWRKY70*-overexpressing (OE) line specifically enriched the genera *Paracoccus*, *Arthrobacter*, and *Rhodococcus* after the leaves of poplar seedlings generated via aseptic tissue culture were inoculated with a consortium consisting of 39 bacterial strains, compared to the wild-type line (28).”.

Lines 877 to 833: “27. Wang W, Wang W, Chen K, Bai X, Zhang X, Li H, Niu B, Jiang J, Liu G. 2024. Effects of transcription factor *PsnWRKY70* on phyllosphere bacterial community of *Populus* infected by *Melampsora laricis-populina*. *Physiol Mol Plant Pathol* 133:102349. <https://doi.org/10.1016/j.pmpp.2024.102349>

28. Wang W, Wang W, Chen K, Bai X, Zhang X, Niu B, Jiang J, Li H, Liu G. 2024. Assembly of phyllosphere bacterial community with *PsnWRKY70* in poplar. *Physiol Mol Plant Pathol* 134:102461. <https://doi.org/10.1016/j.pmpp.2024.102461>”.

I see that there is no Data Availability paragraph at the end of the Methods section. We require that you comply with ASMs Data Policy in order for a manuscript to be published.

We thank editor for pointing this out. We have supplemented “**Data Availability**” at the end of the “**MATERIALS AND METHODS**” section in the revised manuscript by introducing the following text changes:

Lines 687 to 694: “**Data Availability** The original contributions presented in the study are included in the article/Supplementary Material. Further inquiries can be directed to the corresponding author. The sequencing data of the 16S rRNA gene survey reported in this paper have been deposited in the NCBI Sequence Read Archive (BioProject ID no. PRJNA1158217, accession nos. SAMN 43532619-SAMN 43532657). GenBank accession number(s) for six antagonistic strains nucleotide sequence(s): PQ376604-PQ376609.”.

Reviewer #1 (Comments for the Author):

The paper entitled "*Populus simonii* × *P. nigra* overexpressing *PsnWRKY70* recruits phyllosphere bacterial strains that inhibit *Alternaria alternata*" was carefully reviewed. In this work, the authors demonstrated that *PsnWRKY70* gene involved in the plant immunity in the recruitment of specific phyllosphere microbiome against the fungal pathogen in *Populus simonii* × *P. nigra*. The authors attempted to reveal the mechanisms underlying this process by using the 16S rRNA gene amplicon sequencing combined with the plant metabolomics analysis and culturomics techniques. Here are some concerns that need to be addressed:

1. Lines 35-40: these two sentences "To address this...that it had stronger robustness to withstand pathogen infection" are too complex to understand. It is suggested that they be rewritten and combined.

We thank reviewer #1 for this valuable advice. We have deleted the text causing the confusions in the revised manuscript:

“To address this, evaluation of the phyllosphere microbiomes of *Populus* showed significant differences between the wild-type and *PsnWRKY70*-overexpressing lines after inoculation with *A. alternata*. The stability and complexity of the phyllosphere bacterial community was affected by *A. alternata* infection, while higher modularity and a positive correlation with the OE line revealed that it had stronger robustness to withstand pathogen infection.”

We have rewritten the following texts in the updated version of the manuscript:

Lines 35 to 45: “Molecular ecological network analysis demonstrated that the stability and complexity of the phyllosphere bacterial community of *Populus* were influenced by *Alternaria alternata* infection. Specifically, compared to the wild-type line, the *PsnWRKY70*-overexpressing (OE) line had a higher average clustering coefficient and modularity. Furthermore, metabolomic analysis revealed that 19 differential metabolites were significantly enriched in the leaves of the OE line. Among these metabolites, coumarin compounds, such as fraxetin-8-O-glucoside (fraxin) and scopoletin-7-O-glucoside (scopolin), significantly promoted the proliferation of the genera *Methylobacterium* and *Achromobacter* with resistance to *A. alternata*. Additionally, these genera also served as connectors in the molecular ecological network of the phyllosphere microbiome of the OE line.”.

2. Lines 120-124: the three objectives proposed in the "Introduction" section, objectives ii) and iii) are somehow overlapped. It is suggested that they be rewritten and think about a new objective iii) here.

We thank reviewer #1 for this rigorous advice. We have rewritten the objectives ii) and iii) of the "Introduction" section in the revised version of the manuscript:

Lines 126 to 131: “ii) determine the relationships among potential key functional taxa within phyllosphere bacterial communities and the key specific metabolites regulated by the *PsnWRKY70* gene, iii) uncover the mechanism by which key specific metabolites regulated by the *PsnWRKY70* gene in leaves recruit the assembly of key phyllosphere biocontrol bacterial species.”.

3. Line 194: what is RMT short for? please provide its full name in the text.

We thank reviewer #1 for pointing this out. The full name of “RMT” is “random matrix theory”. We have replaced “RMT” with “random matrix theory” in the updated version of the manuscript by introducing the following text changes:

Lines 233 to 234: “we constructed random matrix theory (RMT)-based networks.”

4. Lines 239-240: Here, 568 phyllosphere bacterial strains were isolated. please provide more information about these phyllosphere bacterial strains in the text or Supplementary Materials, such as, How much did they cover the taxa listed in the sequencing data?

We thank reviewer #1 for this question. We have recorrected that the number of phyllosphere bacterial strains isolated was 567 strains. The detail information of the 567 isolated phyllosphere bacterial strains affiliated with 68 genera has been listed in Supplementary Table 8. The high-quality phyllosphere bacterial sequences from all samples were assigned to 236 identified genera. Among them, 31 genera detected were identical to the annotation of 567 isolated bacterial strains. The relative abundance of these 31 genera accounted for 16.28% of the total relative abundance of the phyllosphere microbiome. We have supplemented the texts in the updated version of the manuscript by introducing the following text changes:

Lines 284 to 290: “A total of 567 phyllosphere bacteria strains were isolated and affiliated with 4 phyla (Proteobacteria, Firmicutes, Actinobacteria, and Bacteroidetes) and 68 genera (Supplementary Table 8). Among them, 31 identified genera were detected in the community composition of the phyllosphere microbiome at the genus level. The relative abundance of these 31 genera accounted for 16.28% of the total relative abundance of microbial species in the phyllosphere microbiome of *Populus* (Supplementary Table 9).”.

5. Lines 263-275: it is suggested that "the 91 differential metabolites" should be listed in the supplementary materials, with the amounts of these compounds in the leaves if them available.

We thank reviewer #1 for this valuable advice. The 91 differential metabolites have been listed in **Supplementary Table 7**. Additionally, the relative amounts of these compounds in the leaves have also been provided in the same table. We have also removed the cumbersome names of the differential metabolites in the updated version of the manuscript:

Lines 330 to 333: “Of the differential metabolites, the following showed increased accumulation in the OE line: 8 phenolic acids, 3 flavonoids, 3 lignans and coumarins, 3 terpenoids, 1 alkaloids, and 1 quinones (**Figure 5B and Supplementary Table 7**).”.

Addition, why only choose the two coumarin compounds, scopolin and fraxin, for the bacterial growth test?

We thank reviewer #1 for pointing this out. In total, 91 differential metabolites were identified, of which 19 were significantly increased in the leaves of the OE line compared to the WT line. Among them, their relative contents of two coumarins (fraxin and scopolin) in the OE line are significantly higher than those in the WT line. Furthermore, numerous studies have reported that the coumarins can influence the composition and assembly of the plant microbiome. We speculated that these two compounds could participate in modulating the assembly of the plant microbiome by influencing the proliferation of microbial species. Therefore, scopolin and fraxin were selected to further determine their impact on the proliferation of six biocontrol bacterial isolates. We have supplemented the following texts in the updated version of the manuscript:

Lines 334 to 338: “Among these, the coumarin compounds may play a role in modulating the community composition of the plant microbiome by influencing the proliferation of microbial species (33,34). Therefore, fraxin (MWSmce025) and scopolin (MWSmce024), which belonged to the coumarin compounds, were selected to further determine their impact on the proliferation of six biocontrol bacterial isolates.”.

Lines 893 to 898: “33. Stassen MJJ, Hsu SH, Pieterse CMJ, Stringlis IA. 2021. Coumarin communication along the microbiome–root–shoot axis. *Trends Plant Sci* 26:169-183. <https://doi.org/10.1016/j.tplants.2020.09.008>

34. Voges MJEEE, Bai Y, Schulze-Lefert P, Sattely ES. 2019. Plant-derived coumarins shape the composition of an Arabidopsis synthetic root microbiome. *Proc Natl Acad Sci U S A* 116:12558-12565. <https://doi.org/10.1073/pnas.1820691116>”.

6. Lines 289-291: there were not experimental evidences for the regulation of the biosynthesis of fraxin and scopolin in the leaves by *PsnWRKY70*. It would be great if the authors will be able to talk about the possible mechanisms of this process in the "Discussion" section.

We thank reviewer #1 for pointing this out. We have supplemented the following texts regarding “the biosynthetic pathway of fraxin and scopolin, as well as how does the *WRKY70* gene regulate the biosynthesis of scopolin” in the revised manuscript by introducing the following text modifications:

Lines 434 to 447: “Coumarin biosynthesis originates from the phenylpropanoid pathway, which facilitates the conversion of phenylalanine to p-coumaroyl-CoA (33). The feruloyl-CoA 6'-hydroxylase1 gene (*F6'HI*), encoding a key enzyme in scopoletin biosynthesis, converts feruloyl CoA to 6'-hydroxy-feruloyl CoA. Subsequently, 6'-hydroxy-feruloyl CoA undergoes isomerization of the side chain and lactonization to synthesize scopoletin (59, 60). Scopoletin is then converted into scopolin through the catalysis of UDP-glucosyltransferase. Scopoletin can also be hydroxylated by scopoletin 8-hydroxylase (*S8H*) to generate fraxetin (61). *NaWRKY70* in *Nicotiana attenuata* is crucial for *A. alternata*-induced production of scopoletin and scopolin by directly binding to and activating the promoter of feruloyl-CoA 6'-hydroxylase 1 (*NaF6'HI*), which encodes a key enzyme in scopoletin and scopolin biosynthesis (62). The root-associated microbiome of the scopoletin biosynthesis mutant *f6'h1* of *Arabidopsis thaliana* has demonstrated that scopoletin selectively affects the rhizosphere microbial community assembly (35).”.

Lines 893 to 895: “33. Stassen MJJ, Hsu SH, Pieterse CMJ, Stringlis IA. 2021. Coumarin communication along the microbiome–root–shoot axis. *Trends Plant Sci* 26:169-183. <https://doi.org/10.1016/j.tplants.2020.09.008>”.

Lines 899 to 902: “35. Stringlis IA, Yu K, Feussner K, de Jonge R, Van Bentum S, Van Verk MC, Berendsen RL, Bakker P, Feussner I, Pieterse CMJ. 2018. *MYB72*-dependent coumarin exudation shapes root microbiome assembly to promote plant health. *Proc Natl Acad Sci U S A* 115: E5213-E5222. <https://doi.org/10.1073/pnas.1722335115>”.

Lines 969 to 981: “59. Kai K, Mizutani M, Kawamura N, Yamamoto R, Tamai M, Yamaguchi H, Sakata K, Shimizu Bi. 2008. Scopoletin is biosynthesized via ortho-hydroxylation of feruloyl CoA by a 2-oxoglutarate-dependent dioxygenase in *Arabidopsis thaliana*. *The Plant J* 55(6):989-999. <https://doi.org/10.1111/j.1365-313x.2008.03568.x>

60. Robe K, IEV, F., Rouached H., Dubos C. 2021. The Coumarins: secondary metabolites playing a primary role in plant nutrition and health. *Trends Plant Sci* 26(3):248-259. <https://doi.org/10.1016/j.tplants.2020.10.008>

61. Tsai H-H, Rodríguez-Celma J, Lan P, Wu Y-C, Vélez-Bermúdez IC, Schmidt W. 2018. Scopoletin 8-hydroxylase-mediated fraxetin production is crucial for iron mobilization. *Plant Physiol* 177(1):194-207.

62. Song N, Wu J, Flors V. 2024. Synergistic induction of phytoalexins in *Nicotiana attenuata* by jasmonate and ethylene signaling mediated by NaWRKY70. *J Exp Bot* 75(3):1063-1080. <https://doi.org/10.1104/pp.18.00178>”.

7. Line 294: the "Discussion" section contains too much redundant information. The text should be reduced by focusing on the novel contribution of this study.

We thank reviewer #1 for this valuable advice. In the "Discussion" section, we have deleted some redundant sentences in the previous version of the manuscript:

Lines 405 to 411: “Plant microbiomes are usually considered an indispensable component of plant health , making important contributions to plant growth and disease resistance (31). The plant genotype can play an important role in determining the structure of phyllosphere microbial communities (32). It is still unclear how the *PsnWRKY70* gene in *Populus* can shape the phyllosphere microbiome composition to defend against *A. alternata*. This finding provides insight for recognizing plant resistance genes that play a vital role in shaping the phyllosphere microbiome assembly in *Populus*.”

Lines 413 to 418: “In this study, the dominant taxa in the phyllosphere bacterial communities of *Populus* were mainly Proteobacteria, Actinobacteria, Firmicutes, and Bacteroidetes, which has been observed in previous studies (3, 34). A number of major bacterial genera, including *Pseudonocardia*, *Pseudomonas*, *Sphingomonas*, and *Sphingobium*, constitute the core phyllosphere microbial taxa in WT and(37).”

Lines 487 to 496: “*WRKYs* function to regulate the biosynthesis of alkaloids, terpenes, lignin and phenylpropanoids, among other compounds (33, 52, 53). In this study, we found a higher abundance of phenolic acids and flavonoids in *Populus* leaves. Phenolic acids and flavonoids are plant major secondary metabolites involved in a variety of biological processes in plants, including acquiring nutrients, defending against pathogens, as signaling molecules in plant–microbe symbiosis, and alleviating damage caused by reactive oxygen species (54-57). However, the

metabolites in *Populus* leaves differed between the WT and OE lines when inoculated with *A. alternata*.”

Lines 506 to 511: “We found a higher abundance of coumarin (fraxin and scopolin) in the leaves of the OE line after inoculation with *A. alternata*, which appeared to be closely related to the assembly of the phyllosphere beneficial microbial community in *Populus* that resists pathogen infection. Further confirmation is needed to validate the correlation between coumarins and these differential species of the phyllosphere bacterial community of the OE line.”

Lines 549 to 554: “It can be inferred that the *PsnWRKY70* gene changes the structure of the phyllosphere bacterial community and recruits some biocontrol bacteria to inhibit *A. alternata* infection. Most of the six isolates, especially *Methylobacterium platani* M7161 and *Achromobacter insuavis* TS460, were significantly enriched using coumarins as nutrients or proliferating agents, suggesting that coumarins in *Populus* leaves.”

Lines 569 to 572: “This agrees with previous studies showing that coumarins stimulate the growth and colonization of probiotic rhizobacteria(59, 61). Thus, we considered that coumarins in *Populus* leaves are potent modulators of the phyllosphere microbiome assembly.”

The authors might consider talking about the potential pathway of the modulation of the biosynthesis of the two coumarin compounds and the mechanisms underlying the responses of the antagonistic bacterial strains to these chemicals. In addition, the authors may consider to cite some recent publications to support their ideas in the Discussion.

We have added the potential pathway of the biosynthesis of the two coumarin compounds in the current version of the manuscript:

Lines 434 to 447: “Coumarin biosynthesis originates from the phenylpropanoid pathway, which facilitates the conversion of phenylalanine to p-coumaroyl-CoA (33). The feruloyl-CoA 6'-hydroxylase1 gene (*F6'HI*), encoding a key enzyme in scopoletin biosynthesis, converts feruloyl CoA to 6'-hydroxy-feruloyl CoA. Subsequently, 6'-hydroxy-feruloyl CoA undergoes isomerization of the side chain and lactonization to synthesize scopoletin (59, 60). Scopoletin is then converted into scopolin through the catalysis of UDP-glucosyltransferase. Scopoletin can also be hydroxylated by scopoletin 8-hydroxylase (*S8H*) to generate fraxetin (61). *NaWRKY70* in *Nicotiana attenuata* is crucial for *A. alternata*-induced production of scopoletin and scopolin by directly binding to and activating the promoter of feruloyl-CoA 6'-hydroxylase 1 (*NaF6'HI*), which encodes a key enzyme in scopoletin and scopolin biosynthesis (62). The root-associated

microbiome of the scopoletin biosynthesis mutant *fb h1* of *Arabidopsis thaliana* has demonstrated that scopoletin selectively affects the rhizosphere microbial community assembly (35).”.

Lines 893 to 895: “33. Stassen MJJ, Hsu SH, Pieterse CMJ, Stringlis IA. 2021. Coumarin communication along the microbiome–root–shoot axis. *Trends Plant Sci* 26:169-183. <https://doi.org/10.1016/j.tplants.2020.09.008>”

Lines 899 to 902: “35. Stringlis IA, Yu K, Feussner K, de Jonge R, Van Bentum S, Van Verk MC, Berendsen RL, Bakker P, Feussner I, Pieterse CMJ. 2018. *MYB72*-dependent coumarin exudation shapes root microbiome assembly to promote plant health. *Proc Natl Acad Sci U S A* 115: E5213-E5222. <https://doi.org/10.1073/pnas.1722335115>”.

Lines 969 to 981: “59. Kai K, Mizutani M, Kawamura N, Yamamoto R, Tamai M, Yamaguchi H, Sakata K, Shimizu Bi. 2008. Scopoletin is biosynthesized via ortho-hydroxylation of feruloyl CoA by a 2-oxoglutarate-dependent dioxygenase in *Arabidopsis thaliana*. *The Plant J* 55(6):989-999. <https://doi.org/10.1111/j.1365-313x.2008.03568.x>

60. Robe K, IEV, F., Rouached H., Dubos C. 2021. The Coumarins: secondary metabolites playing a primary role in plant nutrition and health. *Trends Plant Sci* 26(3):248-259. <https://doi.org/10.1016/j.tplants.2020.10.008>

61. Tsai H-H, Rodríguez-Celma J, Lan P, Wu Y-C, Vélez-Bermúdez IC, Schmidt W. 2018. Scopoletin 8-hydroxylase-mediated fraxetin production is crucial for iron mobilization. *Plant Physiol* 177(1):194-207.

62. Song N, Wu J, Flors V. 2024. Synergistic induction of phytoalexins in *Nicotiana attenuata* by jasmonate and ethylene signaling mediated by NaWRKY70. *J Exp Bot* 75(3):1063-1080. <https://doi.org/10.1104/pp.18.00178>”.

We have also supplemented the response mechanisms of the antagonistic bacterial strains to the two coumarin compounds in the updated version of the manuscript as following:

Lines 467 to 476: “Considering the OE line with resistance against *A. alternata* infection and coumarin synthesis in its leaves, coumarins may emerge as potential regulators of the phyllosphere microbiome composition. Most of the six isolates with inhibitory effects against *A. alternata* were significantly enriched in the presence of coumarins. These genera have previously been found to utilize coumarins as a carbon source for growth (70, 71). Other studies have also shown that coumarins can stimulate the growth, chemotaxis, and colonization of probiotic rhizobacteria (35, 72). Thus, we considered that coumarins as nutrients or proliferating agents can be potent

modulators of the phyllosphere microbiome assembly to enhance the niche competition of beneficial microbial species.”.

Lines 899 to 902: “35. Stringlis IA, Yu K, Feussner K, de Jonge R, Van Bentum S, Van Verk MC, Berendsen RL, Bakker P, Feussner I, Pieterse CMJ. 2018. *MYB72*-dependent coumarin exudation shapes root microbiome assembly to promote plant health. *Proc Natl Acad Sci U S A* 115: E5213-E5222. <https://doi.org/10.1073/pnas.1722335115>”.

Lines 1008 to 1016: “70. Guan S, Ji C, Zhou T, Li J, Ma Q, Niu T. 2008. Aflatoxin B1 degradation by *Stenotrophomonas maltophilia* and other microbes selected using coumarin medium. *Int J Mol Sci* 9(8):1489-1503. <https://doi.org/10.3390/ijms9081489>

71. Limaye A, Liu J-R. 2024. Screening and characterization of a *Chryseobacterium timonianum* strain with *Aflatoxin B1* removal ability. *Microb Physiol* 34(1):182-196. <https://doi.org/10.1159/000540803>

72. Mendes R, Kruijt M, Bruijn I, Dekkers E, Voort M, Schneider J HM. 2011. Deciphering the rhizosphere microbiome for disease-suppressive bacteria. *Science* 322(6033):1097-1100. <https://doi.org/10.1126/science.1203980>”.

8. The "Results" section of this manuscript has not been described sufficiently. In particular, in "Effect of key differential metabolites on the growth of six antagonistic strains" (Line 251).

We thank reviewer #1 for pointing this out. We have supplemented principal component analysis of phyllosphere metabolites from the WT and OE lines of *Populus* inoculated with *A. alternata*. Additionally, we provided an explanation for the use of fraxin and scopolin to culture the antagonistic strains, as well as why the concentrations of 20 μ M, 50 μ M, and 500 μ M of these two compounds were used for the bacterial-growth test. We have also described in detail the effect of fraxin or scopolin on the proliferation of six antagonistic isolates. These contents were added to the updated version of the manuscript:

Lines 322 to 326: “Principal component analysis (PCA) revealed a significant difference in leaf metabolites between the WT and OE lines inoculated with *A. alternata* (**Figure S5**). The first two principal components together explained 78.7% of phyllosphere metabolite variance. A distinct separation between the WT and OE lines was noted across the first principal co-ordinate.”.

Lines 334 to 355: “Among these, the coumarin compounds may play a role in modulating the community composition of the plant microbiome by influencing the proliferation of microbial species (33, 34). Therefore, fraxin (MWSmce025) and scopolin (MWSmce024), which belonged

to the coumarin compounds, were selected to further determine their impact on the proliferation of six biocontrol bacterial isolates. The scopolin and fraxin contents were significantly increased in the leaves of the OE line inoculated with *A. alternata* compared to the WT line (**Figure 6A**). To determine whether the *PsnWRKY70* gene directly regulated coumarins to construct the key discriminative taxa in the phyllosphere, a correlation analysis was conducted between the relative abundances of the genera from the six aforementioned bacterial isolates with the antagonistic activity against *A. alternata* in the phyllosphere microbial communities and the fraxin and scopolin contents in the leaves of the WT and OE lines inoculated with *A. alternata* (**Figure S6**). The relative abundances of these six bacterial genera displayed a significant positive correlation with the fraxin and scopolin contents.

To further validate the impact of coumarins on the six antagonistic bacterial isolates, these bacterial strains were cultured with the compounds fraxin and scopolin. Based on the role of scopoletin concentrations in shaping the assembly of the rhizosphere microbial community (35) as well as the scopoletin and scopolin concentrations accumulated in tobacco (36), the concentrations of 20, 50, and 500 μM were set to analyze the effects of scopolin and fraxin on the proliferation of the six biocontrol strains. Most of the bacterial strains tested exhibited growth promotion in the presence of fraxin or scopoline (**Figure 6B**).”.

Lines 893 to 905: “33. Stassen MJJ, Hsu SH, Pieterse CMJ, Stringlis IA. 2021. Coumarin communication along the microbiome–root–shoot axis. *Trends Plant Sci* 26:169-183. <https://doi.org/10.1016/j.tplants.2020.09.008>

34. Voges MJEEE, Bai Y, Schulze-Lefert P, Sattely ES. 2019. Plant-derived coumarins shape the composition of an *Arabidopsis* synthetic root microbiome. *Proc Natl Acad Sci U S A* 116:12558-12565. <https://doi.org/10.1073/pnas.1820691116>

35. Stringlis IA, Yu K, Feussner K, de Jonge R, Van Bentum S, Van Verk MC, Berendsen RL, Bakker P, Feussner I, Pieterse CMJ. 2018. *MYB72*-dependent coumarin exudation shapes root microbiome assembly to promote plant health. *Proc Natl Acad Sci U S A* 115: E5213-E5222. <https://doi.org/10.1073/pnas.1722335115>

36. Xu Z, Zhang S, Wu J. 2023. *NaWRKY3* is a master transcriptional regulator of the defense network against brown spot disease in wild tobacco. *J Exp Bot* 74:4169-4188. <https://doi.org/10.1093/jxb/erad142>”.

Lines 354 to 367: “Most of the bacterial strains tested exhibited growth promotion in the presence of fraxin or scopolin (**Figure 6B**). Both *M. platani* M7161 and *A. insuavis* TS460 used fraxin and scopolin (20 μ M, 200 μ M, 500 μ M) for the proliferation (**Figure 6B**), and these two bacterial genera acted as the connectors in the molecular ecological network of the phyllosphere microbiome of the OE line (**Supplementary Table 5**). Fraxin showed a promoting effect on the growth of *C. soldanellicola* FM524, while having no significant effect on the growth of *P. marcusii* M764. However, only the concentration of 500 μ M fraxin promoted the growth of *F. glaciei* MM675. Besides the concentration of 200 μ M scopolin, scopolin mainly exhibited a promoting effect on the growth of *C. soldanellicola* FM524 and *P. marcusii* M764. Nevertheless, only the concentration of 500 μ M scopolin promoted the proliferation of *F. glaciei* MM675. Furthermore, only *B. vesicularis* M4220 was almost completely inhibited, indicating that the compounds tested could help the active selection of phyllosphere bacteria in *Populus* (**Figure 6B**).”.

We have also provided a detailed description of other research results in the “Results” section in the updated version of the manuscript:

Lines 139 to 153: “**Expression analysis of *PsnWRKY70* in *Populus* leaves**

To confirm the impact of non-inoculation and inoculation with *Alternaria alternata* on the expression of the *PsnWRKY70* gene in the wild-type (WT) and *PsnWRKY70*-overexpressing (OE) lines of *Populus*, the expression levels of the *PsnWRKY70* gene in these two lines were detected by qRT-PCR. The expression level of the *PsnWRKY70* gene in the OE line was significantly higher than that in the WT line during the period from 0 to 36 h, regardless of inoculation with *A. alternata* (**Figure S1A and B**). When not inoculated with *A. alternata*, the expression level of the *PsnWRKY70* gene in the OE line was 1.56–1.90 times that of the WT line (**Figure S1A**). After inoculation with *A. alternata*, the expression level of the *PsnWRKY70* gene in the OE line was 1.98–16.23 times that of the WT line (**Figure S1B**). There was a significant increase in the expression of the *PsnWRKY70* gene in the OE line compared to WT after inoculation with *A. alternata*. This indicates that *A. alternata* infection could significantly upregulate the *PsnWRKY70* expression level in the OE line of *Populus*.”.

Lines 175 to 180: “Compared to non-inoculated plants, the relative abundances of the phylum Firmicutes and [Thermi] in the phyllosphere bacterial community from the WT line significantly declined after infection with *A. alternata*, while a significantly higher abundance of phylum

Proteobacteria was observed along with a significantly lower abundance of the phyla Actinobacteria and Firmicutes in the OE line (**Figure S3B and C**).”.

Lines 188 to 198: “A significantly higher abundance of the genera *Sphingomonas*, *Paracoccus*, and *Sphingobium* characterized the phyllosphere bacterial community from the OE line inoculated with *A. alternata*, while those in the WT line only had a higher abundance of the genera *Pseudonocardia* and *Rickettsia* (**Figure 1C and S4A**). After inoculation with *A. alternata*, the relative abundances of the genera *Pseudonocardia* and *Rickettsia* in the phyllosphere bacterial community from the WT line showed an obvious increase, compared to non-inoculated plants (**Figure S4B**). In the OE line, significantly higher abundances of the genera *Pseudomonas* and *Sphingobium* were observed in samples inoculated with *A. alternata* compared to non-inoculated samples (**Figure S4C**).”.

Lines 209 to 213: “There were no significant differences in the α -diversity index in the phyllosphere microbial community of the OE line between the *A. alternata*-inoculated and non-inoculated treatments; however, the four indices were significantly decreased in the WT line inoculated with *A. alternata* compared to the non-inoculated WT line (**Figure 2C**).”.

Lines 225 to 228: “Furthermore, the phyllosphere microbial communities of both WT ($R^2 = 0.38$, $P = 0.001$) and OE ($R^2 = 0.39$, $P = 0.001$) line inoculated with *A. alternata* showed significant deviations along the positive axis of the first principal co-ordinate, compared to the non-inoculated WT and OE lines (**Figure 2C and D**).”.

Lines 234 to 243: “The molecular ecological networks of the WT and OE lines inoculated with *A. alternata* were characterized by a lower number of total nodes and links and a lower average degree, but a higher R^2 of the power - law, average path distance, and modularity, compared to non-inoculated lines. The empirical networks of the phyllosphere microbiome from two genotypes for both the *A. alternata*-inoculated and non-inoculated treatments had a greater average path distance and higher modularity than their corresponding randomized networks, indicating small-world behavior and modular features (32). However, the empirical network from the WT line inoculated with *A. alternata* had lower modularity compared to that of the non - inoculated line, suggesting that the modular feature was destroyed.”

Lines 284 to 297: “A total of 567 phyllosphere bacteria strains were isolated and affiliated with 4 phyla (Proteobacteria, Firmicutes, Actinobacteria, and Bacteroidetes) and 68 genera (**Supplementary Table 8**). Among them, 31 identified genera were detected in the community

composition of the phyllosphere microbiome at the genus level. The relative abundance of these 31 genera accounted for 16.28% of the total relative abundance of microbial species in the phyllosphere microbiome of *Populus* (**Supplementary Table 9**). Among the 567 phyllosphere bacteria isolates, the genera *Microbacterium*, *Arthrobacter*, and *Frigoribacterium* had relatively high proportions, accounting for 20.63%, 13.40%, and 9.35%, respectively (**Supplementary Table 8**).

Among these isolates, 18 isolates, belonging to the genera *Brevundimonas* (8), *Methylobacterium* (1), *Paracoccus* (2), *Chryseobacterium* (4), *Flavobacterium* (1), and *Achromobacter* (2), were identical to the significantly enriched discriminative genera enriched in the OE line inoculated with *A. alternata* (**Supplementary Table 8 and 11**).”.

Lines 304 to 310: “*F. glaciei* strain MM675 demonstrated a strong inhibitory effect with an inhibition rate of 57.64%, while *B. vesicularis* strain M4220, *A. insuavis* strain TS460, and *M. platani* strain M7161 showed moderate inhibitory effects with inhibition rates of 38.54%, 34.03%, and 27.28%, respectively. Additionally, *P. marcusii* strain M764 and *C. soldanellicola* strain FM524 had the weaker inhibitory effect with an inhibition rate of 18.06% and 13.89% (**Figure 4B and Supplementary Table 10**).”.

In addition, Is there a clear correlation between the contents of the two compounds and the relative abundances of the ASVs representing these biocontrol strains? Please provide more details.

We thank reviewer #1 for this valuable advice. We have added the correlation analysis between the relative contents of fraxin and scopolin and the relative abundance of the genera representing six biocontrol strains (**Figure S6**). We have added the following text to describe the results of Figure S6 and its legend in the updated version of this manuscript:

Lines 340 to 347: “To determine whether the *PsnWRKY70* gene directly regulated coumarins to construct the key discriminative taxa in the phyllosphere, a correlation analysis was conducted between the relative abundances of the genera from the six aforementioned bacterial isolates with the antagonistic activity against *A. alternata* in the phyllosphere microbial communities and the fraxin and scopolin contents in the leaves of the WT and OE lines inoculated with *A. alternata* (**Figure S6**). The relative abundances of these six bacterial genera displayed a significant positive correlation with the fraxin and scopolin contents.”.

Figure S6

Figure S6 Correlation between the relative abundance of the genera from the six biocontrol bacteria isolates and the contents of fraxin (A) and scopolin (B) in the leaves of the WT and OE lines inoculated with *A. alternata*. Shading represents the 95% confidence interval.

9. Figure 6, why did the authors use 20 μM , 50 μM and 500 μM of the two coumarin compounds for the bacterial growth test? The authors may think about providing rationale behind the decision. We thank reviewer #1 for this this valuable question. Previous studies have reported that the contents of scopoletin and scopolin in tobacco leaves were approximately 30 $\mu\text{g/g}$ after inoculation with *A. alternata* (Xu et al., 2023). The concentration was converted into a concentration of around

80 μM in a solution system. Other researchers have also reported that the concentration of scopolin in the root exudates of *Arabidopsis thaliana* was about 300 ng/mL, while the concentration of scopolin in the root was approximately 3000 ng/mg (Stringlis et al. 2023). The concentration was converted into concentrations of approximately 0.85 μM and 8.5 mM, respectively. Thus, we set the concentrations of scopolin at 20 μM , 50 μM , and 500 μM . Previous study investigated the effects of scopoletin (a precursor of scopolin) on bacterial colonization of root surfaces. These studies utilized scopoletin concentrations ranging from 100 μM to 2 mM (Stringlis et al. 2023). Therefore, we referenced these concentrations and selected 20 μM , 50 μM , and 500 μM to explore the effects of scopolin and fraxin on the proliferation of the six biocontrol strains. We have supplemented the following text in the updated version of the manuscript:

Lines 340 to 354: “To determine whether the *PsnWRKY70* gene directly regulated coumarins to construct the key discriminative taxa in the phyllosphere, a correlation analysis was conducted between the relative abundances of the genera from the six aforementioned bacterial isolates with the antagonistic activity against *A. alternata* in the phyllosphere microbial communities and the fraxin and scopolin contents in the leaves of the WT and OE lines inoculated with *A. alternata* (Figure S6). The relative abundances of these six bacterial genera displayed a significant positive correlation with the fraxin and scopolin contents.

To further validate the impact of coumarins on the six antagonistic bacterial isolates, these bacterial strains were cultured with the compounds fraxin and scopolin. Based on the role of scopoletin concentrations in shaping the assembly of the rhizosphere microbial community (35) as well as the scopoletin and scopolin concentrations accumulated in tobacco (36), the concentrations of 20, 50, and 500 μM were set to analyze the effects of scopolin and fraxin on the proliferation of the six biocontrol strains.”.

Lines 899 to 905: “35. Stringlis IA, Yu K, Feussner K, de Jonge R, Van Bentum S, Van Verk MC, Berendsen RL, Bakker P, Feussner I, Pieterse CMJ. 2018. *MYB72*-dependent coumarin exudation shapes root microbiome assembly to promote plant health. *Proc Natl Acad Sci U S A* 115: E5213-E5222. <https://doi.org/10.1073/pnas.1722335115>

36. Xu Z, Zhang S, Wu J. 2023. *NaWRKY3* is a master transcriptional regulator of the defense network against brown spot disease in wild tobacco. *J Exp Bot* 74:4169-4188. <https://doi.org/10.1093/jxb/erad142>”.

Reviewer #2 (Comments for the Author):

This manuscript aimed to reveal how the metabolites regulated by the *PsnWRKY70* gene of *Populus* triggered the phyllosphere microbiome to defend the infection of *Alternaria alternata* by using the multi-omics technologies. The authors found that the phyllosphere microbiome of the wild-type and *PsnWRKY70*-overexpressing lines showed significant difference when being challenged by *A. alternata*, and that, specifically, the stability and complexity of the bacterial communities was influenced by *A. alternata*. The authors also found that the growth of two isolates belonging to the genera *Methylobacterium* and *Achromobacter*, respectively, was significantly promoted by the coumarin compounds, fraxin and scopolin enriched in the OE line, and that both strains were able to inhibit the growth of *A. alternata*. The results shown in this manuscript are interesting, which expand the knowledge of the functionality of gene *PsnWRKY70* in resistance to phytopathogens. Detailed comments are listed as follows:

Major points:

1. L280 to 281: The authors identified quite a few chemicals accumulated in the OE line, why were only fraxin and scopolin used to culture the antagonistic strains? Please give the reason(s) for choosing these two compounds in the text.

We thank reviewer #2 for pointing this out. Their relative contents of two coumarins (fraxin and scopolin) were significantly increased in the leaves of the OE line than those in the wild-type lines. Furthermore, numerous studies have reported that the coumarins can influence the composition of the plant microbiome. We speculated that these two compounds could participate in modulating the assembly of the plant microbiome by influencing the proliferation of microbial species. Therefore, scopolin and fraxin were selected to further determine their impact on the proliferation of six biocontrol bacterial isolates. We have supplemented the following texts in the updated version of the manuscript:

Lines 334 to 338: “Among these, the coumarin compounds may play a role in modulating the community composition of the plant microbiome by influencing the proliferation of microbial species (33,34). Therefore, fraxin (MWSmce025) and scopolin (MWSmce024), which belonged to the coumarin compounds, were selected to further determine their impact on the proliferation of six biocontrol bacterial isolates.”.

Lines 893 to 898: “33. Stassen MJJ, Hsu SH, Pieterse CMJ, Stringlis IA. 2021. Coumarin communication along the microbiome–root–shoot axis. *Trends Plant Sci* 26:169-183. <https://doi.org/10.1016/j.tplants.2020.09.008>

34. Voges MJEEE, Bai Y, Schulze-Lefert P, Sattely ES. 2019. Plant-derived coumarins shape the composition of an *Arabidopsis* synthetic root microbiome. *Proc Natl Acad Sci U S A* 116:12558-12565. <https://doi.org/10.1073/pnas.1820691116>”.

2. L133 to L171: It is interesting that the compositions and diversities of the phyllosphere bacterial communities was influenced by the inoculation of *A. alternata*. It would be more informative if the authors add some data of the comparison of the phyllosphere microbiota of the healthy plants and the ones inoculated with *A. alternata* regardless of the plant genotypes in the first two parts of the RESULTS and update the Figures 1 and 2.

We thank reviewer #2 for this valuable advice. We have supplemented the information regarding the significant change in the community composition of the phyllosphere microbiomes of the WT line (or the OE line) between the *A. alternata*-inoculated and non-inoculated treatments at the phylum or genus level, respectively. We have redrawn Figure S3 and Figure S4, although we didn't update Figure 1. We have supplemented the following texts this in the updated version of the manuscript:

Lines 175 to 180: “Compared to non-inoculated plants, the relative abundances of the phylum Firmicutes and [Thermi] in the phyllosphere bacterial community from the WT line significantly declined after infection with *A. alternata*, while a significantly higher abundance of phylum Proteobacteria was observed along with a significantly lower abundance of the phyla Actinobacteria and Firmicutes in the OE line (**Figure S3B and C**).”.

Lines 188 to 198: “A significantly higher abundance of the genera *Sphingomonas*, *Paracoccus*, and *Sphingobium* characterized the phyllosphere bacterial community from the OE line inoculated with *A. alternata*, while those in the WT line only had a higher abundance of the genera *Pseudonocardia* and *Rickettsia* (**Figure 1B and S4A**). After inoculation with *A. alternata*, the relative abundances of the genera *Pseudonocardia* and *Rickettsia* in the phyllosphere bacterial community from the WT line showed an obvious increase, compared to non-inoculated plants (**Figure S4B**). In the OE line, significantly higher abundances of the genera *Pseudomonas* and *Sphingobium* were observed in samples inoculated with *A. alternata* compared to non-inoculated samples (**Figure S4C**).” .

Figure S3

Figure S3 Comparison of the relative abundance of the dominant phyla in phyllosphere microbial communities between the OE and WT lines of *Populus* inoculated with *Alternaria alternata* (A) Comparison of the relative abundance of the dominant phyla in the WT (B) or OE (C) lines that were non-inoculated and inoculated with *A. alternata*.

Significant differences at the phyla level were analyzed using a two-sided Welch's t-test at a *P*-value of 0.05.

Figure S4

Figure S4 Comparison of the relative abundance of the dominant genera in phyllosphere microbial communities between the OE and WT lines of *Populus* inoculated with *Alternaria alternata*. (A) Comparison of the relative abundance of the dominant genera in the WT (B) or OE (C) lines that were non-inoculated and inoculated with *A. alternata*.

Significant differences at genera level were analyzed using a two-sided Welch's t-test at a *P*-value of 0.05.

We have also added the analyses of the α - and β - diversity in the phyllosphere microbial community of the WT line (or the OE line) between the *A. alternata*-inoculated and non-inoculated treatments. We have supplemented the following texts this in the updated version of the manuscript:

Lines 209 to 213: “There were no significant differences in the α -diversity index in the phyllosphere microbial community of the OE line between the *A. alternata*-inoculated and non-inoculated treatments; however, the four indices were significantly decreased in the WT line inoculated with *A. alternata* compared to the non-inoculated WT line (**Figure 2C**).”.

Lines 225 to 228: “Furthermore, the phyllosphere microbial communities of both WT ($R^2 = 0.38$, $P = 0.001$) and OE ($R^2 = 0.39$, $P = 0.001$) inoculated with *A. alternata* showed significant deviations along the positive axis of the first principal co-ordinate, compared to the non-inoculated WT and OE lines (**Figure 2C and D**).”.

Figure 2

Lines 722 to 735: “**Figure 2 Alpha and principal co-ordinates analysis (PCoA) of phyllosphere microbial communities between *PsnWRKY70*- overexpressing (OE) and wild-type (WT) lines of *Populus*.**

(A) Four diversity indices and significant difference analysis of phyllosphere microbial communities among *Populus* genotypes that were non-inoculated or inoculated with *Alternaria alternata*. * $P < 0.05$; ** $P < 0.01$; *** $P < 0.001$; ns indicates no significant differences. (B) PCoA of the phyllosphere microbial communities between the WT and OE lines that were non-inoculated or inoculated with *A. alternata*. (C) Four diversity indices and significant difference analysis of phyllosphere microbial communities between WT or OE lines that were non-inoculated and inoculated with *A. alternata* respectively. * $P < 0.05$; ** $P < 0.01$; *** $P < 0.001$; ns indicates no significant differences. (D) PCoA of the phyllosphere microbial communities in WT or OE lines that were non-inoculated and inoculated with *A. alternata*.”.

3. The DISCUSSION of this manuscript is definitely too long. The text of the network analysis will need to be reduced. How *A. alternata* infection influences the phyllosphere microbiome and the mechanisms of the suppressive effects of the six biocontrol strains against *A. alternata* are good points to discuss.

We thank reviewer #2 for this question. We have deleted the redundant sentences in the network analysis in the revised manuscript:

Lines 446 to 448: “the phyllosphere bacterial network of the OE line had a higher average clustering coefficient and shorter average path length than for the WT line.”

Lines 451 to 454: “The phyllosphere microbiomes in the WT line not inoculated with *A. alternata* had a significantly higher number of total links and nodes compared to those inoculated with the pathogen.”

Lines 458 to 461: “The lower topological properties in the co-occurrence network of the phyllosphere bacterial community from the WT line inoculated with *A. alternata* also revealed that the bacterial species in the co-occurrence were less affected by the interactions.”

Lines 472 to 474: “After inoculation with *A. alternata*, the stability of the microbial community was broken, and the bacterial community members tended to compete with each other.”

Lines 478 to 479: “Plants with resistant genotypes frequently have a more complex phyllosphere microbial community network (49).”

We have supplemented the following texts regarding “how *A. alternata* infection influences the phyllosphere microbiome” in the current version of the manuscript:

Lines 393 to 396: “A previous study found that when the relative abundance of *Alternaria* was too high, the relative abundance of beneficial bacteria was unlikely to return to a normal level. The phyllosphere microbiomes were unable to regain homeostasis, leading to leaf damage (47).”.

We have also supplemented the antibacterial mechanisms of the six biocontrol strains against *A. alternata* in the updated version of the manuscript as following:

Lines 461 to 467: “Many beneficial functions of these antagonistic bacterial species have been reported, including the secretion of antimicrobial substances, the production of enzymatic activities associated with fungal cell wall degradation (64), the increase in antioxidant enzyme activity of plants (65-67), the triggering of the expression of plant defense genes (68), and plant growth promotion (69). This could potentially serve as a way to help *Populus* resist *A. alternata* infection.”.

Lines 936 to 938: “47. Guo M, Hu J, Jiang C, Zhang Y, Wang H, Zhang X, Hsiang T, Shi C, Wang Q, Wang F. 2024. Response of microbial communities in the tobacco phyllosphere under the stress of validamycin. *Front Microbiol* 14: 1328179. <https://doi.org/10.3389/fmicb.2023.1328179>”.

Lines 985 to 1007: “64. Carrion VJ, Perez-Jaramillo J, Cordovez V, Tracanna V, de Hollander M, Ruiz-Buck D, Mendes LW, van Ijcken WFJ, Gomez-Exposito R, Elsayed SS, Mohanraju P, Arifah A, van der Oost J, Paulson JN, Mendes R, van Wezel GP, Medema MH, Raaijmakers JM. 2019. Pathogen-induced activation of disease-suppressive functions in the endophytic root microbiome. *Science* 366(6465):606-612. <https://doi.org/10.1126/science.aaw9285>

65. Madhaiyan M, Suresh Reddy BV, Anandham R, Senthilkumar M, Poonguzhali S, Sundaram SP, Sa T. 2006. Plant growth-promoting methylobacterium induces defense responses in groundnut (*Arachis hypogaea* L.) compared with rot pathogens. *Curr Microbiol* 53(4):270-276. <https://doi.org/10.1007/s00284-005-0452-9>

66. Attia MS, El-Sayyad GS, Abd Elkodous M, El-Batal AI. 2020. The effective antagonistic potential of plant growth-promoting rhizobacteria against *Alternaria solani*-causing early blight disease in tomato plant. *Sci Hortic* 266:109289. <https://doi.org/10.1016/j.scienta.2020.109289>

67. Faridha Begum I, Mohankumar R, Jeevan M, Ramani K. 2016. GC-MS analysis of bio-active molecules derived from paracoccus pantotrophus FMR19 and the antimicrobial activity against

bacterial pathogens and MDROs. Indian J Microbiol 56(4):426-432. <https://doi.org/10.1007/s12088-016-0609-1>

68. Kumar M, Charishma K, Sahu KP, Sheoran N, Patel A, Kundu A, Kumar A. 2021. Rice leaf associated *Chryseobacterium* species: An untapped antagonistic flavobacterium displays volatile mediated suppression of rice blast disease. Biol Control 161:104703. <https://doi.org/10.1016/j.biocontrol.2021.104703>

69. Sun Z, Yang L, Zhang L, Han M. 2018. An investigation of *Panax ginseng* Meyer growth promotion and the biocontrol potential of antagonistic bacteria against ginseng black spot. J Ginseng Res 42(3):304-311. <https://doi.org/10.1016/j.jgr.2017.03.012>”.

Minor points:

1. L444 to L446: Please provide some parameters of the growth conditions in the greenhouse, for example, temperature, humidity, light and etc.

We thank reviewer #2 for pointing this out. We have added the parameters of the poplar growth conditions in the greenhouse in the updated version of this manuscript as follows:

Lines 521 to 524: “These branches were cut into 15 cm-long brachyblasts and then inoculated with *Alternaria alternata* after 30 days of growth in the greenhouse (16-/8-h light/dark, 25/20°C, and 50–60% relative humidity) (24)”.

Lines 868 to 870: “24. Wang W, Bai X, Chen K, Gu C, Yu Q, Jiang J, Liu G. 2022. Role of *PsnWRKY70* in regulatory network response to infection with *Alternaria alternata* (Fr.) Keissl in *Populus*. Int J Mol Sci 23:7537. <https://doi.org/10.3390/ijms23147537>”.

2. L459 to L462: Please note that the number of the samples for the metabolome analysis is inconsistent with that listed in the supplementary material.

We thank reviewer #2 for pointing this out. We have rectified the number of samples of the widely targeted metabolome in the updated version of this manuscript as following:

Lines 550 to 553: “Leaf samples from the WT and OE lines inoculated with *A. alternata* were collected for widely targeted metabolomics analysis. Each biological replicate sample was composed of 15 leaves pooled from 3 plants, and three independent biological replicates were performed.”.

3. L544 to L545: Please provide the results of the PCA analysis on the metabolites in the leaves of OE and WT lines of poplar.

We very much thank reviewer #2 for this very important advice. We have supplemented PCA analysis to reveal a significant difference in the leaf metabolites between the WT and OE lines inoculated with *A. alternata* (**Figure S5**) in the text of the revised manuscript:

Lines 322 to 326: “Principal component analysis (PCA) revealed a significant difference in phyllosphere metabolites between the WT and OE lines inoculated with *A. alternata* (**Figure S5**). The first two principal components together explained 78.7% of phyllosphere metabolite variance. A distinct separation between the WT and OE lines was noted across the first principal coordinate.”.

Figure S5

Figure S5 Principal component analysis of phyllosphere metabolites from the WT and OE lines of *Populus* inoculated with *A. alternata*.

4. L281 and L427: Is it scopoline or scopolin? Please unitize the spelling of this word through the text.

We thank reviewer #2 for pointing this out. The correct spelling is “scopolin”. We have conducted a comprehensive review of the entire manuscript and rectified all spelling errors in the text of the revised manuscript:

Line 48, Line 336, Line 500, Line 650, Line 758: “scopoline” has been corrected to “scopolin”.

5. L680 to L685: Please indicate the meaning of the asterisks shown in Figures S2 and S3 in their legends.

We thank reviewer #2 for pointing this out. We have redrawn the Figures S2 and S3 using STMP software (v2.1.3) and have updated them to Figures S3 and S4. We have replaced asterisks with the *P*-values in the updated version of the manuscript:

Figure S3

Figure S3 Comparison of the relative abundance of the dominant phyla in phyllosphere microbial communities between the OE and WT lines of *Populus* inoculated with *Alternaria alternata* (A) Comparison of the relative abundance of the dominant phyla in the WT (B) or OE (C) lines that were non-inoculated and inoculated with *A. alternata*.

Significant differences at the phyla level were analyzed using a two-sided Welch's t-test at a *P*-value of 0.05.

Figure S4

Figure S4 Comparison of the relative abundance of the dominant genera in phyllosphere microbial communities between the OE and WT lines of *Populus* inoculated with *Alternaria alternata*. (A) Comparison of the relative abundance of the dominant genera in the WT (B) or OE (C) lines that were non-inoculated and inoculated with *A. alternata*.

Significant differences at genera level were analyzed using a two-sided Welch's t-test at a *P*-value of 0.05.

6. Please clarify the criteria for dividing the topological roles of nodes in MATERIAL AND METHODS, according to the within module connectivity (Z_i) and among module connectivity (P_i).

We thank reviewer #2 for this valuable advice. In the "MATERIAL AND METHODS" section, we have added the criteria for dividing the topological nodes into the four regions based on the values of Z_i and P_i in the updated version of the manuscript:

Lines 676 to 678: "network hubs ($Z_i > 2.5$, $P_i > 0.62$), module hubs ($Z_i > 2.5$, $P_i \leq 0.62$), connectors ($P_i > 0.62$, $Z_i \leq 2.5$), and peripherals ($Z_i \leq 2.5$, $P_i \leq 0.62$)".

7. L231: Some data, for example, the inhibition rate of the six antagonistic bacterial strains against *A. alternata* and/or the radii of the colonies of *A. alternata* co-cultured with these biocontrol strains should be described in the text.

We thank reviewer #2 for this valuable advice. We have added the inhibition rate of the six antagonistic bacterial strains against *A. alternata* in the updated version of the manuscript:

Lines 304 to 310: "*F. glaciei* strain MM675 demonstrated a strong inhibitory effect with an inhibition rate of 57.64%, while *B. vesicularis* strain M4220, *A. insuavis* strain TS460, and *M. platani* strain M7161 showed moderate inhibitory effects with inhibition rates of 38.54%, 34.03%, and 27.28%, respectively. Additionally, *P. marcusii* strain M764 and *C. soldanellicola* strain FM524 had the weaker inhibitory effect with an inhibition rate of 18.06% and 13.89% (**Figure 4B and Supplementary Table 10**).".

8. L700: In Figure 2B, the values of R^2 should be given.

We thank reviewer #2 for this valuable advice. We have added the value of R^2 in the updated Figure 2B in the text of the revised manuscript:

Figure 2

We have added the following text to describe the result in the updated version of this manuscript:

Lines 219 to 222: “PERMANOVA analysis revealed a stronger distinction in phyllosphere microbial communities between OE and WT lines that were non-inoculated ($R^2 = 0.09$, $P = 0.017$) and inoculated ($R^2 = 0.53$, $P = 0.001$) with *A. alternata* (**Figure 2B**).”.

Lines 225 to 228: “Furthermore, the phyllosphere microbial communities of both WT ($R^2 = 0.38$, $P = 0.001$) and OE ($R^2 = 0.39$, $P = 0.001$) lines inoculated with *A. alternata* showed significant deviations along the positive axis of the first principal co-ordinate, compared to the non-inoculated WT and OE lines (**Figure 2C and D**).”.

9. It would be great if the authors will ask an English native speaker to improve the language.

We thank reviewer #2 for this valuable advice. We have asked native English senior experts in this field to polish revised manuscript.

Reviewer #3 (Comments for the Author):

The authors investigate the role of *PsnWRKY70* gene overexpression in hybrid *Populus* strains in response to *Alternaria alternata* fungus infection, focusing on its impact on phyllosphere microbiome assembly and leaf metabolite composition.

To explore this, they compare the phyllosphere microbiomes of overexpression (OE) and wild-type (WT) *Populus* strains before and after pathogen inoculation. They identify bacterial taxa associated with pathogen growth inhibition and determine which metabolites promote these beneficial bacteria. Their findings reveal that the phyllosphere microbiome of OE strains differs from WT strains post-inoculation and using experiments find several bacterial strains exhibiting pathogen-inhibiting properties. Additionally, they confirm that two metabolites, fraxin and scopolin, enhance the growth of these protective bacteria.

Although the exact molecular link between *PsnWRKY70* overexpression and production of specific metabolites is not clear, this study nicely integrates bacterial amplicon sequencing, metabolomics, and experimental validation to highlight and confirm the importance of some potential players in pathogen resistance in OE *Populus* strains. Some technical details in the manuscript need clarification, but overall, I find it a valuable contribution to the literature on fungal resistance mechanisms.

General comments

1. One of my comments is that authors should explain better how OE poplar was constructed,

We thank reviewer #3 for this rigorous advice. The construction of the poplar *PsnWRKY70*-overexpression line by our team was previously published in *Tree Physiology* under the title “*Populus simonii* × *Populus nigra* WRKY70 is involved in salt stress and leaf blight disease responses” (Zhao et al., 2017).

Hui Zhao, Jing Jiang, Kailong Li and Guifeng Liu. *Populus simonii* × *Populus nigra* WRKY70 is involved in salt stress and leaf blight disease responses. *Tree Physiology*. 2017, 37, 827–844.

The construction process of the poplar *PsnWRKY70*-overexpression overexpression line is as follows:

(1) To construct the OE vector, a pair of gene-specific primers, *PsnWRKY70*-F and *PsnWRKY70*-R were designed based on the *Populus trichocarpa* Potri.016G137900 transcript. Then, these

primers were used along with *P. simonii* × *P. nigra* cDNA template to amplify the full length of *PsnWRKY70* ORF. The amplicons were then purified and inserted into pGWB5 vector (**Response Figure 1A**) using pENTR Directional TOPO Cloning Kit and Gateway LR Clonase II enzyme mix (Invitrogen) (Hartley et al., 2000; Zhu et al., 2009). The CaMV 35S promoter- derived pGWB5-*PsnWRKY70*-GFP OE vector (**Response Figure 1B**) was then transferred into *A. tumefaciens* EHA105 cells by electroporation.

(2) OE vectors were transferred into *P. simonii* × *P. nigra* using the *A. tumefaciens*-mediated leaf disc transformation method (Tepfer, 1984; Valvekens et al., 1988; Yevtushenko and Misra, 2010): sterile and wounded *P. simonii* × *P. nigra* leaves were soaked in *A. tumefaciens* solution for 5-10min. The infected leaves were co-cultured on Murashige and Skoog (MS) basal medium (PhytoTechnology Laboratories, Shawnee Mission, KS, USA) supplemented with 0.5 mg l⁻¹ 6-benzylaminopurine (BAP) (Sigma-Aldrich) and 0.1 mg l⁻¹ 1-naphthaleneacetic acid (NAA) (Sigma-Aldrich) in darkness for 2 days. Then, the leaves were transferred to MS medium supplemented with 0.5mg l⁻¹ BAP, 0.1 mg l⁻¹ NAA, 50 mg l⁻¹ kanamycin (Sigma-Aldrich) and 200 mg l⁻¹ cephalosporin (Sigma-Aldrich), incubated at 25 °C for 3-7 weeks until kanamycin-resistant buds appeared. When the resistant buds grew to 3-5 cm, they were transferred to half-strength MS medium supplemented with 20 mg l⁻¹ indole-3-butyric acid (Sigma-Aldrich), and 100 mg l⁻¹ cephalosporin for rooting. Subsequently, the rooting seedlings of *PsnWRKY70* transgenic line and WT line were transferred to soil and cultured for 25 days. The plant growth chamber was maintained at 25 °C ± 2 °C with a 16/8 h photoperiod and a light intensity of 1000 to 1500 lux.

Response Figure 1 Generation and confirmation of *PsnWRKY70* transgenic *P. simonii* × *P. nigra* lines.

(A) Schematic diagram of *pGWB5* empty vector. (B) Schematic diagram of *pGWB5-PsnWRKY70*: GFP vector. C. Acquisition of *PsnWRKY70* transgenic lines.

We have added the following text to the “**MATERIALS AND METHODS**” section in the updated version of this manuscript:

Lines 513 to 520: “**The transgenic *PsnWRKY70*-overexpressing (OE) lines of poplar used in this study were previously described (74). The open reading frame (ORF) of *PsnWRKY70* was amplified from the cDNA of *P. simonii* × *P. nigra* based on the sequence information of *Populus trichocarpa* (Potri.016 G137900). The CaMV 35S promoter-derived *pGWB5-PsnWRKY70*-GFP OE vector was constructed. OE vectors were transferred into *P. simonii* × *P. nigra* to obtain OE lines using the *Agrobacterium. tumefaciens*-mediated leaf-disc transformation method (74).”.**

and how the gene is overexpressed. For example, it is not clear if the gene is overexpressed continuously and by how much compared to WT, and if there are any other molecular differences between OE and WT strains. Do authors know of any other genes that are overexpressed? Authors should specify these details to make sure that gene overexpression is the only factor driving

differences in OE strain. Authors should also show or cite previous results demonstrating that gene expression differs between WT and OE, after fungal inoculation.

We thank reviewer #3 for this valuable advice. The overexpression of PsnWRKY70 transcription factor is bound to affect the changes in the expression levels of downstream genes. In the previous study, we investigated differentially expressed genes regulated by *PsnWRKY70* during disease resistance in poplar by RNA-seq analysis. We also identified direct genome-wide target genes of *PsnWRKY70* using DAP-seq. The key direct target genes regulated by *PsnWRKY70* were verified in vivo by CHIP-PCR and qRT-PCR. These researches were previously published in *International Journal of Molecular Sciences* under the title “Role of PsnWRKY70 in Regulatory Network Response to Infection with *Alternaria alternata* (Fr.) Keissl in *Populus*” (Wang et al., 2022). In the OE line, even if other genes were expressed, they were regulated by the *PsnWRKY70* gene.

Wang W., Bai X.D., Chen K., Gu C. R., Yu Q.B., Jiang J., Liu G.F. Role of PsnWRKY70 in Regulatory Network Response to Infection with *Alternaria alternata* (Fr.) Keissl in *Populus*. *Int. J. Mol. Sci.* 2022, 23, 7537.

However, this study aimed to determine the relationships among potential key functional taxa within phyllosphere bacterial communities and the key specific metabolites regulated by the *PsnWRKY70* gene. It further revealed the mechanism through which the key specific metabolites, regulated by the *PsnWRKY70* gene in leaves, recruit the assembly of key phyllosphere biocontrol bacterial species. Similar research has been published in *Journal of Agricultural and Food Chemistry* under the title “Sultr1;2-Mediated Recruitment of Selenium-Oxidizing Bacteria Promotes Plant Selenium Uptake” (Lei et al., 2025).

Lei Z, Wang HX, Zhang H, et al. Sultr1;2-Mediated Recruitment of Selenium-Oxidizing Bacteria Promotes Plant Selenium Uptake. *Journal of Agricultural and Food Chemistry*. 2025, DOI: 10.1021/acs.jafc.5c01540

We have provided the expression levels of *PsnWRKY70* gene detected in the OE and WT lines both non-inoculated and inoculated with *A. alternata* by qRT-PCR (**Figure S1**). We have added the following texts to describe **Figure S1** and detection method in the updated version of this manuscript:

Lines 139 to 153: “**Expression analysis of *PsnWRKY70* in *Populus* leaves**

To confirm the impact of non-inoculation and inoculation with *Alternaria alternata* on the expression of the *PsnWRKY70* gene in the wild-type (WT) and *PsnWRKY70*-overexpressing (OE)

lines of *Populus*, the expression levels of the *PsnWRKY70* gene in these two lines were detected by qRT-PCR. The expression level of the *PsnWRKY70* gene in the OE line was significantly higher than that in the WT line during the period from 0 to 36 h, regardless of inoculation with *A. alternata* (**Figure S1A and B**). When not inoculated with *A. alternata*, the expression level of the *PsnWRKY70* gene in the OE line was 1.56–1.90 times that of the WT line (**Figure S1A**). After inoculation with *A. alternata*, the expression level of the *PsnWRKY70* gene in the OE line was 1.98–16.23 times that of the WT line (**Figure S1B**). There was a significant increase in the expression of the *PsnWRKY70* gene in the OE line compared to WT after inoculation with *A. alternata*. This indicates that *A. alternata* infection could significantly upregulate the *PsnWRKY70* expression level in the OE line of *Populus*.”.

Lines 524 to 534: “The expression levels of the *PsnWRKY70* gene were determined in leaf samples of the WT and OE lines both non-inoculated and inoculated with *A. alternata* (0, 6, 12, 24, and 36 h) via qRT-PCR. Total RNA of each sample was extracted using a Universal Plant Total RNA Extraction Kit (BioTeke Corporation, Beijing, China), and cDNA was separately synthesized using the Toyobo Reverse Transcription Kit (ReverTra Ace® qPCR RT Master Mix with gDNA Remover) (Toyobo, Osaka, Japan). qRT-PCR was performed on an ABI-7500 quantitative PCR instrument using Toyobo SYBR® Green Real-time PCR Master Mix Plus (Toyobo, Osaka, Japan) (74). 18S was used as the internal control to normalize the expression levels of the *PsnWRKY70* gene. The relative expression levels of the target gene were calculated using the $2^{-\Delta\Delta CT}$ method as described previously (74).”

Lines 1019 to 1021: “74. Zhao H, Jiang J, Li K, Liu G. 2017. *Populus simonii* x *Populus nigra* WRKY70 is involved in salt stress and leaf blight disease responses. *Tree Physiol* 37(6):827-844. <https://doi.org/10.1093/treephys/tpx020>”

Figure S1

Figure S1 Expression levels of the *PsnWRKY70* gene in the WT and OE lines of *Populus* measured after inoculation with H₂O (A) and *Alternaria alternata* (B) over a period of 0–36 h.

2. I found it hard to follow the section on how specific bacteria were selected for experimental inhibition tests. Some important details are missing, for example, authors refer to a phylogenetic method to subset bacterial strains enriched in OE after fungus inoculation, but the analysis is not described anywhere in the methods, and furthermore it's not written how many strains were found to be enriched, and how many strains were selected for the inhibition test. Please provide all necessary details.

We thank reviewer #3 for pointing this out. Based on LEfSe analysis, the potential discriminative taxa of phyllosphere microbial communities between OE and WT lines were distinguished after inoculation with *A. alternaria*. We identified that 29 discriminative taxa were significantly enriched in the OE line. Among them, 21 taxa were annotated to the identified genus, including *Sphingomonas*, *Sphingobium*, *Azospirillum*, *Rhizobium*, *Alkanindiges*, *Paracoccus*, *Skermanella*, *Methylotenera*, *Achromobacter*, *Flavisolibacter*, *Flavobacterium*, *Devosia*, *Brevundimonas*, *Janthinobacterium*, *Chryseobacterium*, *Mycoplana*, *Novosphingobium*, *Sphingopyxis*, *Methylobacterium*, *Dietzia* and *Phenylobacterium* (**Supplementary Table 11**). Among the 567 isolated bacterial strains from the leaves, six genera (namely *Brevundimonas*, *Methylobacterium*, *Paracoccus*, *Chryseobacterium*, *Flavobacterium*, and *Achromobacter*) belong to the significantly enriched discriminative taxa in the OE line after inoculation with *A. alternaria*. Among these isolates, 18 isolates belonging to the genera *Brevundimonas* (8), *Methylobacterium* (1),

Paracoccus (2), *Chryseobacterium* (4), *Flavobacterium* (1), and *Achromobacter* (2), were identical to the significantly enriched discriminative genera enriched in the OE line inoculated with *A. alternata* (**Supplementary Table 11**). Among these six genera, bacterial isolates exhibiting the highest similarity to the ASVs with the highest relative abundance in each genus of the phyllosphere microbial communities of *Populus* was selected for the inhibition tests (**Supplementary Table 12**). Therefore, the strains *Brevundimonas vesicularis* M4220, *Methylobacterium platani* M7161, *Paracoccus marcusii* M764, *Chryseobacterium soldanellicola* FM524, *Flavobacterium glaciei* MM675, and *Achromobacter insuavis* TS460 were selected to evaluate their antagonistic activity against *A. alternata* in Petri plate antagonism assays. We have supplemented the following texts in the updated version of this manuscript:

Lines 284 to 304: “A total of 567 phyllosphere bacteria strains were isolated and affiliated with 4 phyla (Proteobacteria, Firmicutes, Actinobacteria, and Bacteroidetes) and 68 genera (**Supplementary Table 8**). Among them, 31 identified genera were detected in the community composition of the phyllosphere microbiome at the genus level. The relative abundance of these 31 genera accounted for 16.28% of the total relative abundance of microbial species in the phyllosphere microbiome of *Populus* (**Supplementary Table 9**). Among the 567 phyllosphere bacteria isolates, the genera *Microbacterium*, *Arthrobacter*, and *Frigoribacterium* had relatively high proportions, accounting for 20.63%, 13.40%, and 9.35%, respectively (**Supplementary Table 8**).

Among these isolates, 18 isolates belonging to the genera *Brevundimonas* (8), *Methylobacterium* (1), *Paracoccus* (2), *Chryseobacterium* (4), *Flavobacterium* (1), and *Achromobacter* (2), were identical to the significantly enriched discriminative genera enriched in the OE line inoculated with *A. alternata* (**Supplementary Table 8 and 11**). Among these six genera, bacterial isolates exhibiting the highest similarity to the ASVs with the highest relative abundance in each genus of the phyllosphere microbial communities of *Populus* was selected for the antagonistic bioassays against *A. alternata* (**Supplementary Table 12**). Therefore, six bacterial isolates, identified as *Brevundimonas vesicularis* M4220, *Methylobacterium platani* M7161, *Paracoccus marcusii* M764, *Chryseobacterium soldanellicola* FM524, *Flavobacterium glaciei* MM675, and *Achromobacter insuavis* TS460, were selected to evaluate their antagonistic activity against *A. alternata* in Petri plate antagonism assays.”.

Minor comments:

1. Page2, Abstract - Abbreviations and names are written for the first time without explanation. Please explain what is OE, and what are fraxin and scopolin.

We thank reviewer #3 for pointing this out. “OE” refers to “the *PsnWRKY70* - overexpressing (OE) line”, while “fraxin” and “scoplin” refer to “fraxetin-8-O-glucoside (fraxin)” and “scopoletin-7-O-glucoside (scopolin)”, respectively. We have supplemented the following texts in the abstract of the revised manuscript:

Line 38: “the *PsnWRKY70* - overexpressing (OE) line”.

Lines 41 to 42: “fraxetin-8-O-glucoside (fraxin) and scopoletin-7-O-glucoside (scopolin)”.

2. Page 7, lines 145-148: this statement should be supported by the tests, eg those shown in Fig. S2 Fig. S2 - Please add what asterisks mean.

We thank reviewer #3 for pointing this out. We have redrawn the Figures S2 and S3 using STMP software (v2.1.3) and have updated them to Figures S3 and S4. Additionally, in both of these figures, we have replaced asterisks with the *P*-values. We have updated a version of Figure S3, S4 and their legend in the text of the revised manuscript:

Figure S3

Figure S3 Comparison of the relative abundance of the dominant phyla in phyllosphere microbial communities between the OE and WT lines of *Populus* inoculated with *Alternaria alternata* (A) Comparison of the relative abundance of the dominant phyla in the WT (B) or OE (C) lines that were non-inoculated and inoculated with *A. alternata*.

Significant differences at the phyla level were analyzed using a two-sided Welch's t-test at a *P*-value of 0.05.

A

B

C

Figure S4

Figure S4 Comparison of the relative abundance of the dominant genera in phyllosphere microbial communities between the OE and WT lines of *Populus* inoculated with *Alternaria alternata*. (A) Comparison of the relative abundance of the dominant genera in the WT (B) or OE (C) lines that were non-inoculated and inoculated with *A. alternata*.

Significant differences at genera level were analyzed using a two-sided Welch's t-test at a *P*-value of 0.05.

We have also added the following text to describe the results regarding **Figure S3** and **S4** in the updated version of this manuscript:

Lines 175 to 180: “Compared to legend -inoculated plants, the relative abundances of the phylum Firmicutes and [Thermi] in the phyllosphere bacterial community from the WT line significantly declined after infection with *A. alternata*, while a significantly higher abundance of phylum Proteobacteria was observed along with a significantly lower abundance of the phyla Actinobacteria and Firmicutes in the OE line (**Figure S3B and C**).” .

Lines 192 to 198: “After inoculation with *A. alternata*, the relative abundances of the genera *Pseudonocardia* and *Rickettsia* in the phyllosphere bacterial community from the WT line showed an obvious increase, compared to non-inoculated plants (**Figure S4B**). In the OE line, significantly higher abundances of the genera *Pseudomonas* and *Sphingobium* were observed in samples inoculated with *A. alternata* compared to non-inoculated samples (**Figure S4C**).”.

3. Page 8, lines 163-165: I find this sentence hard to follow. What authors mean by: "the genotype distribution characteristics of host plants after inoculation with *A. alternata*" - please rewrite or clarify.

We thank reviewer #3 for pointing this out. We have rewritten the composition of the phyllosphere microbiomes at the genus level in the WT and OE lines both non-inoculated and inoculated with *A. alternata*. We have deleted this sentence in the previous version of the manuscript:

Lines 163 to 165: “the genotype distribution characteristics of host plants after inoculation with *A. alternata*” is inappropriate in the text, and we have removed it.”

We have revised the following texts in the updated version of the manuscript:

Lines 188 to 198: “A significantly higher abundance of the genera *Sphingomonas*, *Paracoccus*, and *Sphingobium* characterized the phyllosphere bacterial community from the OE line inoculated with *A. alternata*, while those in the WT line only had a higher abundance of the genera *Pseudonocardia* and *Rickettsia* (**Figure 1B and S4A**). After inoculation with *A. alternata*, the

relative abundances of the genera *Pseudonocardia* and *Rickettsia* in the phyllosphere bacterial community from the WT line showed an obvious increase, compared to non-inoculated plants (**Figure S4B**). In the OE line, significantly higher abundances of the genera *Pseudomonas* and *Sphingobium* were observed in samples inoculated with *A. alternata* compared to non-inoculated samples (**Figure S4C**).”.

4. Page 8, line 179-181: A suggestion - it looks to me that when comparing inoculated with non-inoculated, diversity decreases after inoculation, but to a smaller degree in OE strains. Authors could compare non-inoculated and inoculated strains to test it.

We thank reviewer #3 for this valuable advice. We have also added the analyses of the α - and β -diversity in the phyllosphere microbial community of the WT line (or the OE line) between the *A. alternata*-inoculated and non-inoculated treatments. We have supplemented the following texts in the updated version of the manuscript:

Lines 209 to 213: “There were no significant differences in the α -diversity index in the phyllosphere microbial community of the OE line between the *A. alternata*-inoculated and non-inoculated treatments; however, the four indices were significantly decreased in the WT line inoculated with *A. alternata* compared to the non-inoculated WT line (**Figure 2C**).”.

Lines 225 to 228: “Furthermore, the phyllosphere microbial communities of both WT ($R^2 = 0.38$, $P = 0.001$) and OE ($R^2 = 0.39$, $P = 0.001$) lines inoculated with *A. alternata* showed significant deviations along the positive axis of the first principal co-ordinate, compared to the non-inoculated WT and OE lines (**Figure 2C and D**).”.

Lines 722 to 735: “**Figure 2 Alpha and principal co-ordinates analysis (PCoA) of phyllosphere microbial communities between *PsnWRKY70*- overexpressing (OE) and wild-type (WT) lines of *Populus*.**

(A) Four diversity indices and significant difference analysis of phyllosphere microbial communities among *Populus* genotypes that were non-inoculated or inoculated with *Alternaria alternata*. * $P < 0.05$; ** $P < 0.01$; *** $P < 0.001$; ns indicates no significant differences. (B) PCoA of the phyllosphere microbial communities between the WT and OE lines that were non-inoculated or inoculated with *A. alternata*. (C) Four diversity indices and significant difference analysis of phyllosphere microbial communities between WT or OE lines that were non-inoculated and inoculated with *A. alternata* respectively. * $P < 0.05$; ** $P < 0.01$; *** $P < 0.001$; ns indicates no

significant differences. (D) PCoA of the phyllosphere microbial communities in WT or OE lines that were non-inoculated and inoculated with *A. alternata*.”

5. Page 9, line 85 - Authors could add R^2 to show that there is indeed a stronger distinction.

We thank reviewer #3 for this valuable advice. We have added R^2 in the updated Figure 2B of the revised manuscript:

Figure 2

We have also added the following text in the updated version of this manuscript:

Lines 219 to 222: “PERMANOVA analysis revealed a stronger distinction in phyllosphere microbial communities between OE and WT lines that were non-inoculated ($R^2 = 0.09$, $P = 0.017$) and inoculated ($R^2 = 0.53$, $P = 0.001$) with *A. alternata* (**Figure 2B**).”.

Lines 225 to 228: “Furthermore, the phyllosphere microbial communities of both WT ($R^2 = 0.38$, $P = 0.001$) and OE ($R^2 = 0.39$, $P = 0.001$) lines inoculated with *A. alternata* showed significant deviations along the positive axis of the first principal co-ordinate, compared to the non-inoculated WT and OE lines (**Figure 2C and D**).”.

6. Page 9, line 195-8: "All empirical networks." Please add a reference to a figure for this sentence.

We thank reviewer #3 for this valuable advice. We have added a reference for this sentence in the text of the revised manuscript:

Lines 237 to 241: “The empirical networks of the phyllosphere microbiome from two genotypes had a greater average path distance and higher modularity than their corresponding randomized networks, indicating small-world behavior and modular features (32)”.

Lines 891-892: “32. Deng Y, Jiang YH, Yang Y, He Z, Luo F, Zhou J. 2012. Molecular ecological network analyses. *BMC Bioinf* 13:113. <https://doi.org/10.1186/1471-2105-13-113>”.

7. Page 9, line 203 - "bacterial network of the OE line" - Please specify here if you mean inoculated or non-inoculated one

We thank reviewer #3 for this valuable advice. "bacterial network of the OE line" refers to “bacterial network of the OE line inoculated with *A. alternata* ". We have clarified the sentence in the text of the revised manuscript:

Lines 248 to 252: “The bacterial network of the OE line inoculated with *A. alternata* had the highest number of total nodes and links, the highest average path distance, and greater average connectivity (avgK) (**Table 1 and Figure 3**), indicating that it has a more complex network, greater network robustness, and higher density of connections compared to that of the WT line.”.

8. Page 10 - lines 209-211 - Are you comparing here networks before and after inoculation, or WT and OE networks? From the table it looks like before inoculation both WT and OE networks are most complex.

We thank reviewer #3 for pointing this out. Indeed, the number of total nodes and links in the molecular ecological networks from the WT and OE lines inoculated with *A. alternata* was lower compared to the non-inoculated lines. Meanwhile, the phyllosphere microbiome from these two genotypes had a greater average path distance and higher modularity after inoculation with *A.*

alternata, indicating small-world behavior and modular features. However, the higher modularity in empirical networks from the WT line inoculated with *A. alternata* was lower than that in their corresponding randomized networks, suggesting that the modular feature was destroyed. This could be attributed to the disturbance in the balance of the phyllosphere microbial community in the WT line by *A. alternata*, which caused the community structure to disintegrate. However, after inoculation with *A. alternata*, the phyllosphere microbiome in the OE line of *Populus* still had a higher modularity, indicating that it contributes to the rapid response of the microbial community to defend against pathogen infection and reduce the external disturbance. These results provide evidence that the *PsnWRKY70* gene in *Populus* can resist pathogen infection by shaping the microbial community network in the phyllosphere. We have supplemented the following the text in the updated version of the manuscript:

Lines 234 to 243: “The molecular ecological networks of the WT and OE lines inoculated with *A. alternata* were characterized by a lower number of total nodes and links and a lower average degree, but a higher R^2 of the power - law, average path distance, and modularity, compared to non-inoculated lines. The empirical networks of the phyllosphere microbiome from two genotypes for both the *A. alternata*-inoculated and non-inoculated treatments had a greater average path distance and higher modularity than their corresponding randomized networks, indicating small-world behavior and modular features (32). However, the empirical network from the WT line inoculated with *A. alternata* had lower modularity compared to that of the non - inoculated line, suggesting that the modular feature was destroyed.”.

9. Page 11 - line 241-243 - Referring to general comments: Please explain how this analysis was done. Also not clear, how many strains were selected for inhibition test.

We thank reviewer #3 for pointing this out. We determined the taxonomic status of 567 bacterial isolates by means of 16S rRNA gene alignment (**Supplementary Table 8**). We identified that 29 discriminative taxa were significantly enriched in the OE line after inoculation with *A. alternaria* based on LEfSe analysis. Among them, 21 taxa were annotated to the identified genus, including *Sphingomonas*, *Sphingobium*, *Azospirillum*, *Rhizobium*, *Alkanindiges*, *Paracoccus*, *Skermanella*, *Methylothera*, *Achromobacter*, *Flavisolibacter*, *Flavobacterium*, *Devosia*, *Brevundimonas*, *Janthinobacterium*, *Chryseobacterium*, *Mycoplana*, *Novosphingobium*, *Sphingopyxis*, *Methylobacterium*, *Dietzia* and *Phenylobacterium* (**Supplementary Table 11**). Among the 567 isolated bacterial strains from the leaves, six genera (namely *Brevundimonas*, *Methylobacterium*,

Paracoccus, *Chryseobacterium*, *Flavobacterium*, and *Achromobacter*) belong to the significantly enriched discriminative taxa in the OE line after inoculation with *A. alternaria*. Among these isolates, 18 isolates belonging to the genera *Brevundimonas* (8), *Methylobacterium* (1), *Paracoccus* (2), *Chryseobacterium* (4), *Flavobacterium* (1), and *Achromobacter* (2), were identical to the significantly enriched discriminative genera enriched in the OE line inoculated with *A. alternata* (**Supplementary Table 11**). Among these six genera, bacterial isolates exhibiting the highest similarity to the ASVs with the highest relative abundance in each genus of the phyllosphere microbial communities of *Populus* was selected for the inhibition tests (**Supplementary Table 12**). Therefore, the strains *Brevundimonas vesicularis* M4220, *Methylobacterium platani* M7161, *Paracoccus marcusii* M764, *Chryseobacterium soldanellicola* FM524, *Flavobacterium glaciei* MM675, and *Achromobacter insuavis* TS460 were selected to evaluate their antagonistic activity against *A. alternata* in Petri plate antagonism assays. We have supplemented the following texts in the updated version of this manuscript:

Lines 284 to 304: “A total of 567 phyllosphere bacteria strains were isolated and affiliated with 4 phyla (Proteobacteria, Firmicutes, Actinobacteria, and Bacteroidetes) and 68 genera (**Supplementary Table 8**). Among them, 31 identified genera were detected in the community composition of the phyllosphere microbiome at the genus level. The relative abundance of these 31 genera accounted for 16.28% of the total relative abundance of microbial species in the phyllosphere microbiome of *Populus* (**Supplementary Table 9**). Among the 567 phyllosphere bacteria isolates, the genera *Microbacterium*, *Arthrobacter*, and *Frigoribacterium* had relatively high proportions, accounting for 20.63%, 13.40%, and 9.35%, respectively (**Supplementary Table 8**).

Among these isolates, 18 isolates belonging to the genera *Brevundimonas* (8), *Methylobacterium* (1), *Paracoccus* (2), *Chryseobacterium* (4), *Flavobacterium* (1), and *Achromobacter* (2), were identical to the significantly enriched discriminative genera enriched in the OE line inoculated with *A. alternata* (**Supplementary Table 8 and 11**). Among these six genera, bacterial isolates exhibiting the highest similarity to the ASVs with the highest relative abundance in each genus of the phyllosphere microbial communities of *Populus* was selected for the antagonistic bioassays against *A. alternata* (**Supplementary Table 12**). Therefore, six bacterial isolates, identified as *Brevundimonas vesicularis* M4220, *Methylobacterium platani* M7161, *Paracoccus marcusii* M764, *Chryseobacterium soldanellicola* FM524, *Flavobacterium glaciei* MM675, and

Achromobacter insuavis TS460, were selected to evaluate their antagonistic activity against *A. alternata* in Petri plate antagonism assays.”.

10. Figure 5A - Please specify what the scale represents.

We thank reviewer #3 for pointing this out. **Figure 5A** showed the percentage of the relative content of metabolites in each category accounting for the total amount from the leaves of WT and OE lines of *Populus* inoculated with *Alternaria alternata*. We have rewritten the legend of **Figure 5A** in the text of updated manuscript:

Lines 754 to 756: “(A) The percentage of the relative content of metabolites in each category accounting for the total amount in the leaves of WT and OE lines of *Populus* inoculated with *Alternaria alternata*.”.

11. Supplementary Table 6 and lines 258-261 - The table does not show which metabolites are enriched. Perhaps you should add a column with enrichment or refer to another figure/result.

We thank reviewer #3 for pointing this out. We have failed to describe these texts clearly. These texts described in lines 258-261 was about Figure 5A. “enrichment” has been replaced with “relative content”, and “Supplementary Table 6” has been replaced with “Figure 5A” in the text of the revised manuscript:

Lines 318 to 321: “The leaves of WT and OE lines exhibited the higher relative content of phenolic acids, flavonoids, and alkaloids, indicating that the types and distribution of metabolites in the leaves of the two *Populus* genotype inoculated with *A. alternata* were generally similar (**Figure 5A**)”.

Re: mSystems01765-24R1 (*Populus simonii* × *P. nigra* overexpressing *PsnWRKY70* recruits phyllosphere bacterial strains that inhibit *Alternaria alternata*)

Dear Dr. Ben Niu:

I have read the manuscript myself and have still a few suggestions to improve its clarity:

>Please Indicate the link between Fig. 4 panels A and B: Why these strains were tested, I guess as reps of the taxa enriched in OE lines inoculated indicate this in panel A maybe even with the % of inhibition?

>Please correct the text for sentences which convey a finalistic view of the hypotheses and results. Here is An example L57: "However, little is known about how the metabolites regulated by the PsnWRKY70 gene trigger the phyllosphere microbiome to defend against foliar pathogens." This sentence implies a will or aim in the expression of the gene (or the plant) which is not scientifically sound. Please consider changing to: "...trigger changes in the phyllosphere microbiome, leading to increase resistance to foliar pathogens."

>These sentences are highlighted in the text (attached pdf) and I have proposed some clarifications in the comments section.

Revision Guidelines

Sincerely,

Juliana Almario
Editor
mSystems

Reviewer #1 (Comments for the Author):

The authors have addressed the concerns I raised.

Reviewer #2 (Comments for the Author):

The authors have made the revision as per my suggestions and comments.

Reviewer #3 (Comments for the Author):

The authors thoroughly revised the manuscript and put a lot of effort into addressing all the reviewers' comments. I am entirely satisfied with the comments and corrections made.

A small correction that authors may want to check, in line 219: "PERMANOVA analysis revealed stronger distinction between lines that were non-inoculated ...". I believe the authors meant here something else, "weaker distinction between lines that were non-inoculated".

Dear Editor,

We thank you for the time and efforts in evaluating our manuscript. We very much appreciate the comments and suggestions provided and have modified the manuscript accordingly. Before providing a detailed point-by-point response we simply summarize the major changes contained in the new version of the manuscript. The major changes in the revised version can be summarized as follows:

- 1) We have strengthened the link between Figure 4A and 4B to clarify the reason why 6 discriminative genera of biocontrol strains enriched in the OE line inoculation with *A. alternata* were selected.
- 2) We have corrected some sentences using scientific wording in the manuscript.
- 3) We have revised the section regarding the PERMANOVA analysis to more clearly distinguish the principal co-ordinate analysis (PCoA) of phyllosphere microbial communities in non - inoculated and *A. alternata* - inoculated OE and WT lines.

In the following pages, we provide a point-by-point response to each of the editor's comments. The original comments are presented in blue font and our responses are in black font and we used red font where we place in the letter the modified text as it appears in the revised manuscript. We have also directly replied to all the comments marked by editor in the text file of the revised manuscript. Beside this response letter, we also attach two comparison files showing the changes we have made on the previous version of text to get the current version of the manuscript. We feel the manuscript has been greatly improved thanks to the thoughtful comments and suggestions you have provided. We hope you will now find our work suitable for publication in mSystems.

Sincerely,

Wei Wang, Weixiong Wang, Di Wu and Ben Niu (for all authors)

Editor (Comments for the Author):

>Please Indicate the link between Fig. 4 panels A and B: Why these strains were tested, I guess as reps of the taxa enriched in OE lines inoculated indicate this in panel A maybe even with the % of inhibition?

We thank editor for this rigorous advice. Plant resistance genes can enhance the plant resistance to pathogen infection by influencing the assembly of the plant's beneficial microbiome(33). Considering that a stronger distinction in phyllosphere microbial communities between OE and WT lines inoculated with *A. alternata*, the significantly discriminative genera enriched in the OE line inoculated with *A. alternata* were used as a reference to screen for antagonistic strains (Figure 4A). Six biocontrol bacterial strains against *A. alternata* from Figure 4B have been marked in Figure 4A.

We have supplemented the texts in the updated version of the manuscript by introducing the following text changes:

Lines 295 to 300: “Plant resistance genes can enhance the plant resistance to pathogen infection by influencing the assembly of the plant's beneficial microbiome(33). Considering that a stronger distinction in phyllosphere microbial communities between OE and WT lines inoculated with *A. alternata*, the significantly discriminative genera enriched in the OE line inoculated with *A. alternata* were used as a reference to screen for antagonistic strains (Figure 4A).”

Lines 778 to 787: “**Figure 4 Linear discriminant analysis effect size (LEfSe) analysis of phyllosphere community composition and isolation of bacterial strains with antagonistic potential against *A. alternata*.**

(A) Discriminant analysis of species differences in phyllosphere community composition between *PsnWRKY70*-overexpressing and wild-type *Populus* lines non-inoculated or inoculated with *A. alternata* based on LEfSe analysis. Six significantly discriminative genera enriched in the OE line were selected for the antagonistic bioassays against *A. alternata* and were marked with blue triangles. (B) In vitro antagonism bioassays against *A. alternata* and the inhibition rate of six antagonistic isolates against *A. alternata*.”

Lines 927 to 930: “33. Yang K, Fu R, Feng H, Jiang G, Finkel O, Sun T, Liu M, Huang

B, Li S, Wang X, Yang T, Wang Y, Wang S, Xu Y, Shen Q, Friman V-P, Jousset A, Wei Z. 2023. *RIN* enhances plant disease resistance via root exudate-mediated assembly of disease-suppressive rhizosphere microbiota. *Mol Plant* 16:1379-1395. <https://doi.org/10.1016/j.molp.2023.08.004>

>Please correct the text for sentences which convey a finalistic view of the hypotheses and results. Here is An example L57: "However, little is known about how the metabolites regulated by the *PsnWRKY70* gene trigger the phyllosphere microbiome to defend against foliar pathogens." This sentence implies a will or aim in the expression of the gene (or the plant) which is not scientifically sound. Please consider changing to: "...trigger changes in the phyllosphere microbiome, leading to increase resistance to foliar pathogens."

We thank editor for this valuable advice. We have rectified the following texts in the updated version of this manuscript as following:

Lines 57 to 59: “However, little is known about how the metabolites regulated by the *PsnWRKY70* gene trigger changes in the phyllosphere microbiome, leading to increase resistance to foliar pathogens.”

L60: “adjusted coumarin synthesis in the leaves”. This sentence is confusing. Consider changing to: exhibited increased coumarin synthesis in the leaves, triggering changes in microbial species central in phyllosphere microbial networks, and leading to increase resistance to *A. alternata* infection.

We thank editor for this question. We have revised the texts in the updated version of the manuscript by introducing the following text changes:

Lines 59 to 63: “Here, the *PsnWRKY70* overexpressing line of *Populus* (*Populus simonii* × *P. nigra*) exhibited increased coumarin synthesis in the leaves, triggering changes in microbial species central in phyllosphere microbial networks, and leading to increase resistance to *A. alternata* infection.”

L62: resist infection

We thank editor for pointing this out. We have deleted the text “resist infection” and rewritten the following texts in the updated version of the manuscript:

Lines 59 to 63: “Here, the *PsnWRKY70* overexpressing line of *Populus* (*Populus simonii* × *P. nigra*) exhibited increased coumarin synthesis in the leaves, triggering changes in microbial species central in phyllosphere microbial networks, and leading to increase resistance to *A. alternata* infection.”

L80-81: “can also control the immune system to shape the phyllosphere microbial community” consider changing to: “can shape phyllosphere microbial communities via their immune system”.

We thank editor for this valuable advice. We have revised the following texts in the updated version of the manuscript:

Lines 80 to 81: “Plant genetic networks can shape phyllosphere microbial communities via their immune system and maintain plant health (16).”

L104-105: “It recruited beneficial microbes to enhance poplar's resistance to

Melampsora laricis – populina” consider changing to: “...beneficial microbes leading to enhanced poplar resistance”.

We thank editor for this question. We have corrected the following texts in the updated version of the manuscript:

Lines 104 to 106: “It recruited beneficial microbes leading to enhance poplar resistance to *Melampsora laricis – populina* (27).”

L114-115: “play a role in controlling the microbial community structure in the phyllosphere to withstand” consider rephrasing to: “...play a role in shaping the ... phyllosphere, leading to limited pathogen infection via the regulation of secondary”.

We thank editor for this valuable advice. We have rectified the following texts in the updated version of this manuscript as following:

Lines 113 to 116: “it remains unknown whether WRKY70 TFs play a role in shaping the microbial community structure in the phyllosphere, leading to limited pathogen infection via the regulation of the secondary metabolites produced by *Populus*.”

L123-124: “to resist infection by *A. alternata*” consider rephrasing to: “triggering resistance to *A. Alternata* infection”.

We thank editor for pointing this out. We have revised the following texts in the updated version of the manuscript:

Lines 121 to 124: “However, no studies have combined leaf metabolomics and phyllosphere bacterial microbiome to better understand the mechanism by which *PsnWRKY70* regulates *Populus* (*Populus simonii* × *P. nigra*) triggering resistance to *A. alternata* infection.”

L156-157: “To confirm whether the *PsnWRKY70* gene regulated the microbiomes to withstand pathogens” consider rephrasing to: “To test whether *PsnWRKY70* modulates leaf microbiome assembly thereby limiting pathogen development”.

We thank editor for this valuable advice. We have amended the texts in the updated version of the manuscript by introducing the following text changes:

Lines 156 to 158: “To test whether the *PsnWRKY70* modulates leaf microbiome assembly thereby limiting pathogen development,”

L340-342: “To determine whether the *PsnWRKY70* gene directly regulated coumarins

to construct the key discriminative taxa in the phyllosphere” consider rephrasing to: “.. gene directly regulated coumarins synthesis thereby influencing the abundance of microbial taxa in the phyllosphere”.

We thank editor for pointing this out. We have revised the following texts in the updated version of the manuscript:

Lines 348 to 350: “To determine whether the *PsnWRKY70* gene directly regulated coumarins synthesis thereby influencing the abundance of microbial taxa in the phyllosphere,”

L366-367: “that the compounds tested could help the active selection of phyllosphere bacteria in *Populus*” consider rephrasing to: “could participate in the active selection of”.

We thank editor for this valuable advice. We have corrected the following texts in the updated version of the manuscript:

Lines 375 to 376: “indicating that the compounds tested could participate in the active selection of phyllosphere bacteria in *Populus* (Figure 6B).”

L371-372: “Plants can defend against pathogen infection and protect plant health by enriching certain beneficial microorganisms”

We thank editor for this question. We have rewritten the following texts in the updated version of the manuscript:

Lines 380 to 381: “The plant genotype can play an important role in regulating certain beneficial phyllosphere microorganisms to defend against pathogen infection (16, 38)”

L378: “to avoid” correct to: “avoiding”.

We thank editor for this valuable advice. We have amended the following texts in the updated version of the manuscript:

Lines 387 to 388: “and produce bioactive molecules to induce stomatal closure avoiding pathogen entry into the apoplast (41).”

L398-399: “the relationships of phyllosphere bacterial populations within the network from the OE line were better connected”. Please clarify this sentence. Bacterial networks exhibited higher connectedness?

We thank editor for pointing this out. We have revised the following texts in the updated

version of the manuscript:

Lines 408 to 410: “the relationships of phyllosphere bacterial populations within the network from the OE line exhibited higher connectedness,”

L405: “disintegrate” consider changing to: “shift significantly”

We thank editor for this question. We have rectified the following texts in the updated version of this manuscript as following:

Lines 415 “which caused the community structure to shift significantly.”

L417-418: “indicating that it may be more inclined to form a” consider changing to: “associated to more stable microbial communities”.

We thank editor for this this valuable question. We have amended the texts in the updated version of the manuscript by introducing the following text changes:

Lines 427 to 429: “indicating that it may be associated to more stable microbial community structure due to the mutualistic relationships within modules (56).”

L420: “can resist” consider changing to: “participates in plant resistance by”

We thank editor for pointing this out. We have rectified the following texts in the updated version of this manuscript as following:

Lines 429 to 431: “These results provide evidence that the *PsnWRKY70* gene in *Populus* participates in plant resistance to pathogen infection by shaping the microbial community network in the phyllosphere.”

L422: “planata” please correct this sentence.

We thank editor for this valuable advice. We have deleted the ambiguous word “planata” in the revised manuscript:

Lines 432: “in vitro antagonistic activity assays”

L425: “to enhance” consider rephrasing to: “potentially enhancing their”

We thank editor for this question. We have corrected the following texts in the updated version of the manuscript:

Lines 436 to 437: “to potentially enhancing their ability to fight pathogens”

L478-479: “vital impacts on the connection of modules in” consider rephrasing: “an important role in network structure”.

We thank editor for this valuable advice. We have revised the texts in the updated

version of the manuscript by introducing the following text changes:

Lines 489-490 “suggesting an important role in network structure (74).”

L482-483: “to modify the phyllosphere microbiome assembly to prevent infection by *A. alternata*” consider changing to: “thereby modifying the phyllosphere microbiome and preventing infection by *A. alternata*”.

We thank editor for this question. We have rectified the following texts in the updated version of this manuscript as following:

Lines 493 to 494: “thereby modifying the phyllosphere microbiome and preventing infection by *A. alternata*.”

Reviewer #3 (Comments for the Author):

A small correction that authors may want to check, in line 219: "PERMANOVA analysis revealed stronger distinction between lines that were non-inoculated ...". I believe the authors meant here something else, "weaker distinction between lines that were non-inoculated".

We very much thank reviewer #3 for this very important advice. We have rewritten the following texts in the updated version of the manuscript:

Lines 220 to 223: “PERMANOVA analysis revealed a stronger distinction in phyllosphere microbial communities between OE and WT lines that were inoculated with *A. alternata* ($R^2 = 0.53$, $P = 0.001$), and a weaker distinction between the non – inoculated OE and WT lines ($R^2 = 0.09$, $P = 0.017$) (Figure 2B).”

Re: mSystems01765-24R2 (*Populus simonii* × *P. nigra* overexpressing *PsnWRKY70* recruits phyllosphere bacterial strains that inhibit *Alternaria alternata*)

Dear Dr. Ben Niu:

Your manuscript has been accepted, and I am forwarding it to the ASM production staff for publication. Your paper will first be checked to make sure all elements meet the technical requirements. ASM staff will contact you if anything needs to be revised before copyediting and production can begin. Otherwise, you will be notified when your proofs are ready to be viewed.

Sincerely,
Juliana Almario
Editor
mSystems